# Cis-regulatory architecture of human ESC-derived hypothalamic neuron differentiation aids in variant-to-gene mapping of relevant complex traits

Matthew C. Pahl[1,12], Claudia A. Doege[2,12], Kenyaita M. Hodge[1,12], Sheridan H. Littleton [1,12], Michelle E. Leonard[1], Sumei Lu[1], Rick Rausch[3], James A. Pippin [1], Maria Caterina De Rosa[3], Alisha Basak[3], Jonathan P. Bradfield[1], Reza K. Hammond[1], Keith Boehm[1], Robert I. Berkowitz[4], Chiara Lasconi[1,5], Chun Su[1], Alessandra Chesi[1,5], Matthew E. Johnson[1], Andrew D. Wells[1,6,7], Benjamin F. Voight [8,9], Rudolph L. Leibel[10], Diana L. Cousminer[1,5,8,11,13] & Struan F. A. Grant [1,4,5,8,13✉]

The hypothalamus regulates metabolic homeostasis by influencing behavior and endocrine systems. Given its role governing key traits, such as body weight and reproductive timing, understanding the genetic regulation of hypothalamic development and function could yield insights into disease pathogenesis. However, given its inaccessibility, studying human hypothalamic gene regulation has proven challenging. To address this gap, we generate a high-resolution chromatin architecture atlas of an established embryonic stem cell derived hypothalamic-like neuron model across three stages of in vitro differentiation. We profile accessible chromatin and identify physical contacts between gene promoters and putative cis-regulatory elements to characterize global regulatory landscape changes during hypothalamic differentiation. Next, we integrate these data with GWAS loci for various complex traits, identifying multiple candidate effector genes. Our results reveal common target genes for these traits, potentially affecting core developmental pathways. Our atlas will enable future efforts to determine hypothalamic mechanisms influencing disease susceptibility.

[1] Center for Spatial and Functional Genomics, Children's Hospital of Philadelphia, Philadelphia, PA 19104, USA. [2] Department of Pathology, Naomi Berrie Diabetes Center, Columbia Stem Cell Initiative, Columbia University Vagelos College of Physicians and Surgeons, New York, NY, USA. [3] Department of Pediatrics, Naomi Berrie Diabetes Center, Columbia Stem Cell Initiative, Columbia University, Vagelos College of Physicians and Surgeons, New York, NY, USA. [4] Department of Pediatrics, The University of Pennsylvania Perelman School of Medicine, Philadelphia, PA 19104, USA. [5] Division of Human Genetics, Children's Hospital of Philadelphia, Philadelphia, PA 19104, USA. [6] Institute for Immunology, Perelman School of Medicine, University of Pennsylvania, Philadelphia, PA, USA. [7] Department of Pathology and Laboratory Medicine, Children's Hospital of Philadelphia, Philadelphia, PA, USA. [8] Department of Genetics, University of Pennsylvania, Philadelphia, PA 19104, USA. [9] Department of Systems Pharmacology and Translational Therapeutics, Perelman School of Medicine, University of Pennsylvania, Philadelphia, PA 19104, USA. [10] Division of Molecular Genetics (Pediatrics) and the Naomi Berrie Diabetes Center, Columbia University Vagelos College of Physicians and Surgeons, New York, NY, USA. [11] Present address: GSK, Human Genetics and Computational Biology, 1250 South Collegeville Road, Collegeville, PA 19426, USA. [12] These authors contributed equally: Matthew C. Pahl, Claudia A. Doege, Kenyaita M. Hodge, Sheridan H. Littleton. [13] These authors jointly supervised this work: Diana L. Cousminer, Struan F.A. Grant. ✉email: grants@email.chop.edu

The hypothalamus is a critical regulator of many physiological functions, including energy homeostasis, reproduction, sleep, and stress[1]. This brain region senses neural and physiological signals, which triggers distinct populations of neurons to release neurotransmitters and peptide neuromodulators to signal the autonomic nervous and endocrine systems[1–3]. Monogenic mutations in key nutrient-sensing hypothalamic genes, such as the leptin and melanocortin 4 receptors, result in obesity through dysregulating the neural circuit involved in controlling hunger and satiety, while mutations impacting gonadotrophin-releasing hormone signaling impair the onset of puberty by disrupting pituitary gland signaling[2,4].

There is a lack of epigenomic data characterizing the genetic regulatory architecture of the developing and mature human hypothalamus, limiting our ability to translate studies into information directly relevant for disease[5]. Recently, improvements in embryonic and induced pluripotent stem cell differentiation strategies[1,5,6] have partially mitigated the need to study human hypothalamic neurons ex vivo. As the precise regulation of hypothalamic development remains poorly understood, differentiating hypothalamic neurons from ESCs provides an opportunity to study these cells and their precursors over time, which could lead to a greater understanding of the development of hypothalamic-governed traits and diseases.

Genome-wide association studies (GWAS) have yielded hundreds of loci statistically associated with phenotypes known to involve hypothalamic function[7–12]. GWAS efforts typically only report single-nucleotide polymorphisms (SNPs) yielding the statistically strongest associations per locus. However, these lead SNPs are not necessarily the causal variants due to the presence of other SNPs in linkage disequilibrium (LD). The majority of GWAS signals reside in noncoding regions of the genome, suggesting that their impact on phenotype is primarily via gene regulation. As cis-acting regulatory elements (cREs), such as enhancers or silencers, can act locally or over large genomic distances, the nearest gene to a GWAS signal may not be the principal effector gene[13–16]. Thus, a major challenge in complex trait genetics is to confidently identify the precise regulatory variant(s) tagged by sentinel SNPs and their corresponding effector target gene(s).

Chromatin conformation approaches to identify SNP-harboring regions that contact effector genes via long-range promoter interactions in various cell and tissue contexts[17–19]. Recently, we combined a suite of techniques to systematically evaluate GWAS signals located in distal elements[20–23]. Together, our integrated "variant-to-gene mapping" approach aims to physically fine-map significant GWAS loci by identifying open proxy SNPs in LD with each given sentinel signal that directly contacts a gene promoter. Assaying relevant cell types in this regard is critical, as promoter architecture varies across cellular identity and developmental stage[17,24,25].

While changes in hypothalamic gene expression during development have been studied[26,27], the corresponding cis-regulatory architecture in hypothalamic neuron differentiation remains largely unexplored. In this study, we use an arsenal of molecular techniques to characterize the genetic architecture of differentiation of embryonic stem cells, first into hypothalamic progenitors (HPs) and then arcuate (ARC) nucleus-like hypothalamic neurons (HN). While the hypothalamus consists of a diverse array of neuronal subtypes, we approached this using bulk sequencing approach on differentiated cells. The term "hypothalamic neurons (HN)" will be used to describe the differentiated cell population composed of a diverse set of differentiated hypothalamic-like neurons, and a small population of non-neuronal cells. Utilizing this model, we subsequently superimpose GWAS findings for relevant traits on these data to implicate critical and novel effector genes, along with their corresponding putative regulatory elements.

## Results

**ESC-derived hypothalamic-like neurons (HN) recapitulate molecular characteristics of the hypothalamus.** We utilized an established protocol to derive ARC HN-like neurons that generate predominantly neurons that express markers such as NPY and POMC (80–95%)[28], and collected cells at three stages of differentiation: pluripotent ESCs, NKX2-1+ hypothalamic progenitors (HPs), and HNs generated from a human ESC line (H9) derived from one female donor. Twelve days were selected as the HP timepoint due to high expression of the neuroprogenitor marker Nestin and the low expression of the neuronal marker Tubulin Beta 3 (*TUBB3*), while day 27 was chosen as HN timepoint due to high TUBB3 and POMC expression[28]. We then profiled global gene expression patterns for these three stages using RNA-seq, chromatin accessibility with ATAC-seq, and chromatin conformation via promoter-focused Capture C to generate a high-resolution atlas of the distal promoter interaction landscape in an in vitro human model of hypothalamic development (Fig. 1a). To assess the reproducibility between replicates (separate differentiations), we performed principal component analysis and pairwise Pearson correlation on the RNA-seq and ATAC-seq datasets. In both cases, the first principal component corresponded to the stage of differentiation and accounted for more than half of the variation (RNA-seq: 52.60%; ATAC-seq: 55.30%) (Supplementary Fig. 1a–d).

To further confirm the molecular congruence of HN differentiation to the in vivo development of HNs, we examined the expression of several marker genes (Fig. 1b and Supplementary Fig. 1e)[26], which were consistent with expectations[28,29]. As a negative control, we examined the expression of the developing telencephalon/forebrain marker *FOXG1*, and which as expected was detected in low levels in HPs and later expressed in HN (Supplementary Fig. 1c), similar to its absence from hypothalamic progenitors and later expression in subsets hypothalamic neurons later in development[26].

In particular, we investigated the expression and promoter interaction landscape of *NKX2-1*, which encodes a transcription factor (TF) critical for hypothalamus specification. *NKX2-1* is expressed in the developing hypothalamus and subsequently becomes restricted to a subset of neurons[30]. *NKX2-1* expression followed a similar pattern during HN differentiation (Fig. 1b). In addition, we observed a distinct change in the accessibility of the *NKX2-1* promoter, concordant with its expression pattern (Fig. 1c), as well as fewer interactions detected as *NKX2-1* expression decreased (Fig. 1d), confirming our detection of expected dynamic changes.

In addition to confirming the expression of known marker genes, we compared the global transcriptomic profile of HNs to the GTeX RNA-seq database, which is derived from primary human tissue samples (Fig. 1e and Supplementary Data 1)[31]. HN gene expression was highly correlated with the hypothalamus (Spearman's rho= 0.719; adjusted $P = 1.14 \times 10^{-14}$). Taken together, these results show that ESC-derived HNs resemble hypothalamus tissue. To supplement the comparison with GTeX, we queried the top 500 expressed genes in HNs against a database of brain region-specific marker genes[32,33], which had been previously been used to verify the identity of iPSC-derived neurons[5]. The strongest cell-type enrichment for the HN gene set was the hypothalamus hypocretinergic neurons (Fisher's Exact test: FDR = $3.863 \times 10^{-04}$) as well as weaker enrichment detected in striatal cholinergic neurons (Fisher's Exact test: FDR = 0.010).

As the differentiated HNs represent multiple hypothalamic neuronal subtypes from the hypothalamus[28,34], bulk sequencing

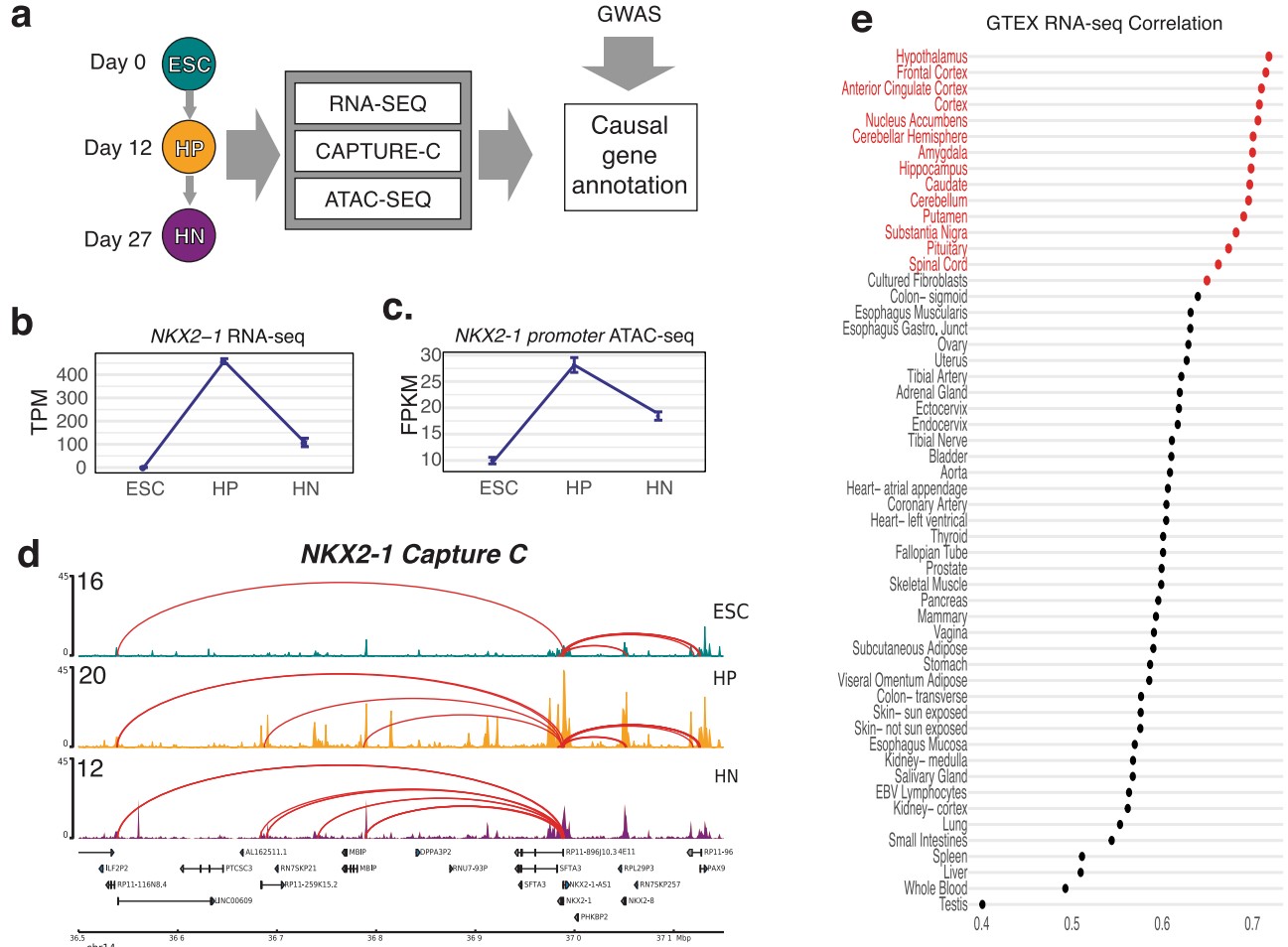

**Fig. 1 An integrative functional genomics approach to model the differentiation of hypothalamic neurons. a** Schematic of the study design. ESC, HP, and HNs were used to generate RNA-seq, ATAC-seq, and Capture C profiles, which we compared to the GWAS signals mined using our variant-to-gene mapping approach. **b** Expression level of NKX2-1 determined by RNA-seq (error bars reflect standard deviation; n = 3). **c** Accessibility change of OCR located in the NKX2-1 promoter over the course of HN differentiation (error bars reflect standard deviation; ESC, HP n = 4; HN n = 6). **d** A 600 kb region around the NKX2-1 gene in ESCs (teal), HPs (purple), and HNs (orange). The peaks track represents the ATAC-seq coverage, where higher peaks depict increased accessibility (open chromatin), and arcs represent significant (Chicago Score >5) interacting regions between the NKX2-1 promoter. **e** Comparison of HN expression profile to median GTeX database, scores give the Spearman correlation coefficient of the top 16,953 genes expressed in both datasets.

approaches are not sufficient to capture the cellular heterogeneity of the tissue. Previous single-cell RNA-seq of neurons differentiated using this protocol have previously been shown to consist of POMC, SST, and AGRP/NPY neural subtypes[34]. While *POMC* and *SST* were detected at high levels, we detected very low levels of *NPY* and *AGRP* (Supplementary Fig. 1c).

**Temporal dynamics of regulation of gene expression and *cis*-regulatory elements during hypothalamic neuron differentiation.** We assessed the temporal profile of gene expression to identify genes with developmental stage-restricted expression during human HN differentiation, with 15,808 genes differentially expressed in at least one stage (Fig. 2a). We assigned these genes to six clusters based on expression patterns during the course of differentiation. Each cluster corresponded to genes specifically enriched or depleted in at least one stage of differentiation (Fig. 2b and Supplementary Fig. 2a, b). Gene Ontology (GO) and REACTOME enrichment analysis of each cluster identified known biological processes related to proliferating progenitor cells and differentiated neurons[35] (Supplementary Fig. 2c, d).

To correlate gene expression changes during HN differentiation with the respective chromatin accessibility and conformation profiles at each stage, we defined the relationship with open chromatin regions (OCRs) using ATAC-seq. We identified a total of 404,691 OCRs in at least one stage. The OCRs were disproportionately located in promoters (−1500/+500 bp TSS) and first introns (Supplementary Fig. 3a), which is characteristic of regulatory elements[36]. We intersected this list with promoter contacts called in our Capture C data (442,779 promoter contacts called in ESCs, 347,919 called in HPs, and 366,062 called in HNs). We then grouped the OCRs into three categories (Fig. 2c): (1) OCRs located within promoter regions annotated as "promoter OCRs"; (2) OCRs with direct promoter contacts determined by Capture C, annotated as "promoter-interacting region (PIR)-OCRs"; and (3) OCRs that could not be assigned to a gene because they did not fit either criterion, annotated as "non-PIR-OCRs". Because they could be annotated to a gene, we considered the sets of 50,952 promoter OCRs and 87,170 PIR-OCRs as putative cREs.

Both the number of cREs per gene (median of three PIR-OCRs in each cell type) and the mean distance between the cRE and the

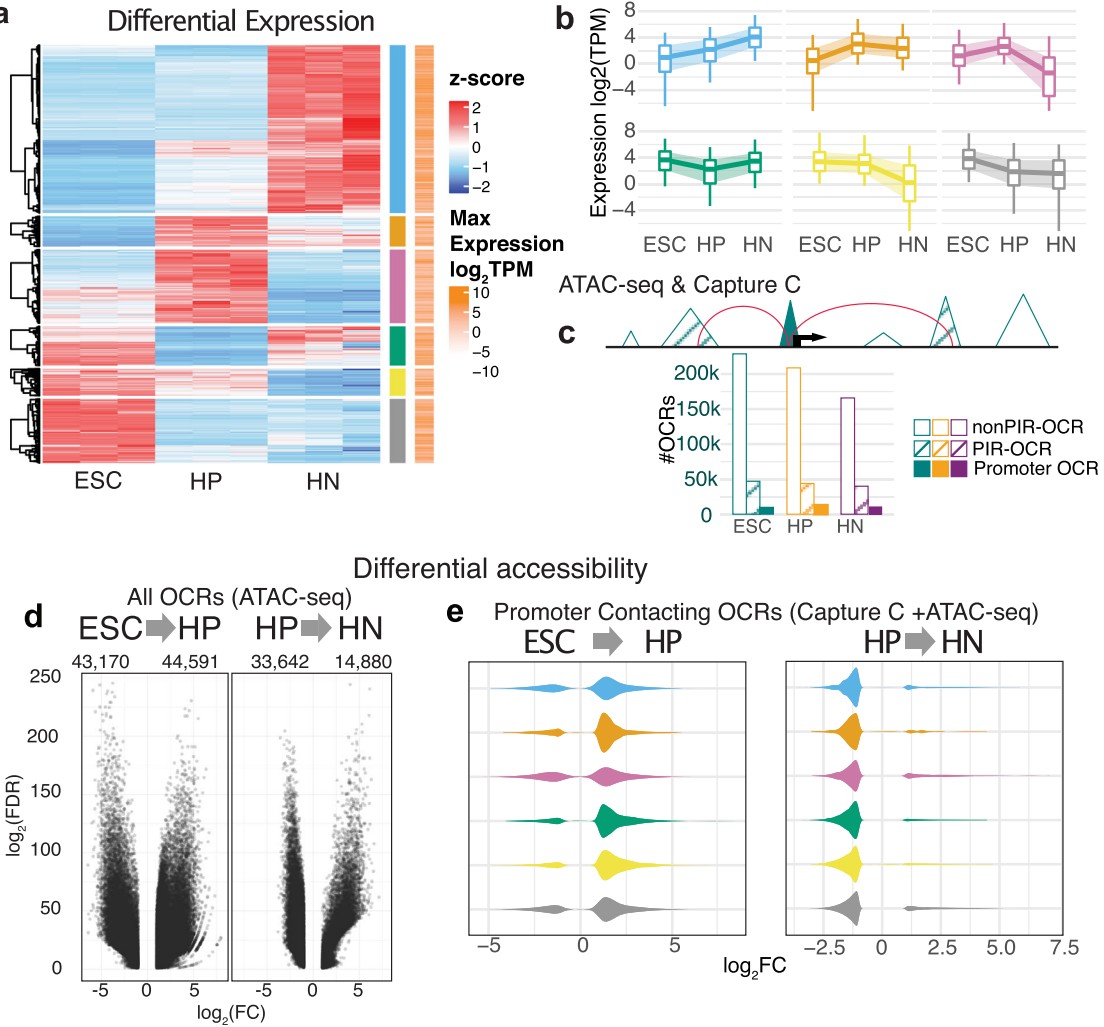

**Fig. 2 Changes in gene expression and chromatin architecture underlying hypothalamic neuron differentiation. a** Heatmap of the scaled expression values (*z*-score) of each RNA-seq sample of ESC, HP, and HN for the 15,808 genes with differential expression in at least one stage of differentiation. Red indicates higher relative expression, while blue indicates lower relative expression. From hierarchical clustering, the genes were assigned to six groups based on their expression pattern pointed to being either specifically enriched or depleted in one condition (blue, orange, pink, green, yellow, gray bars). The log-transformed condition-wise max TPM value for each gene is plotted where white indicates lower expression and orange indicates higher expression compared to other genes. **b** The global average of expression values (TPM) of the genes in each cluster. The central line of each boxplot represents the median, with edges representing the 25 and 75 percentiles, and whiskers represent the 5 and 95 percentiles. **c** Sets of OCRs were assigned based on location: promoter OCRs (solid), PIR-OCRs (hashed), and non-PIR-OCRs the remaining OCRs that were not annotated to a gene (promoter OCR-ESCs: 9796, HPs: 12,962, HNs: 10,241; PIR-OCR- ESCs: 46,968, HPs: 43,860, HNs: 39,942; non-PIR-OCR ESC: 228,747, HP: 208,763, HN: 165,860). Bottom: the distribution of number OCRs annotated to each set per cell type. **d** Volcano plot depicting the global genome-wide significant differentially accessible OCRs in the transition from ESC to HP (left) and HP to HN (right). **e** Distribution of chromatin accessibility fold change of cRE annotated to DE gene clusters (**a**).

promoter were decreased in HPs compared to ESCs or HNs (Supplementary Fig. 3b, c), reflecting fewer long-range interactions detected at this stage (Supplementary Fig. 3d, e). We also observed a trend for genes with higher expression interacting with more cREs (Kruskal–Wallis test: *P* value $<2.2 \times 10^{-16}$) (Supplementary Fig. 3f), which is in line with reports for other neuronal[37] and immune cells[23].

We then compared chromatin accessibility across the three stages, and identified 87,761 differentially accessible regions from ESCs to HPs (43,170 more closed; 44,591 more open) and 48,522 differentially accessible regions from HPs to HNs (33,642 more closed; 14,880 more open) (Fig. 2d and Supplementary Fig. 3d). The genome-wide decrease in open chromatin as cells advanced in differentiation occurs in other developing tissues[38]. The subset of cREs contacting differentially expressed genes showed a net

increase in accessibility as ESCs differentiated into HPs, and subsequently decreased as HPs advanced to HNs, regardless of the gene expression pattern defined by our clustering analysis (Fig. 2e, a). This trend suggests that regulatory elements driving the gene expression changes specifying undifferentiated progenitors to a hypothalamic cell fate in this cellular model are primarily first established by opening of selected cREs followed by a more global pruning of contacts upon differentiation to neurons.

To provide context on some of the open accessible regions, we compared the set of gene-connected cRE in HN to a previous epigenomic study that compared the enhancer landscape (ATAC-seq and H3K27ac ChIP-seq) of sorted leptin positive and negative hypothalamic neurons from mice[39]. We found an enriched overlap of the mouse hypothalamic neuron ATAC-seq and H3K27ac peaks with the HN cRE (Supplementary Fig. 4a). In

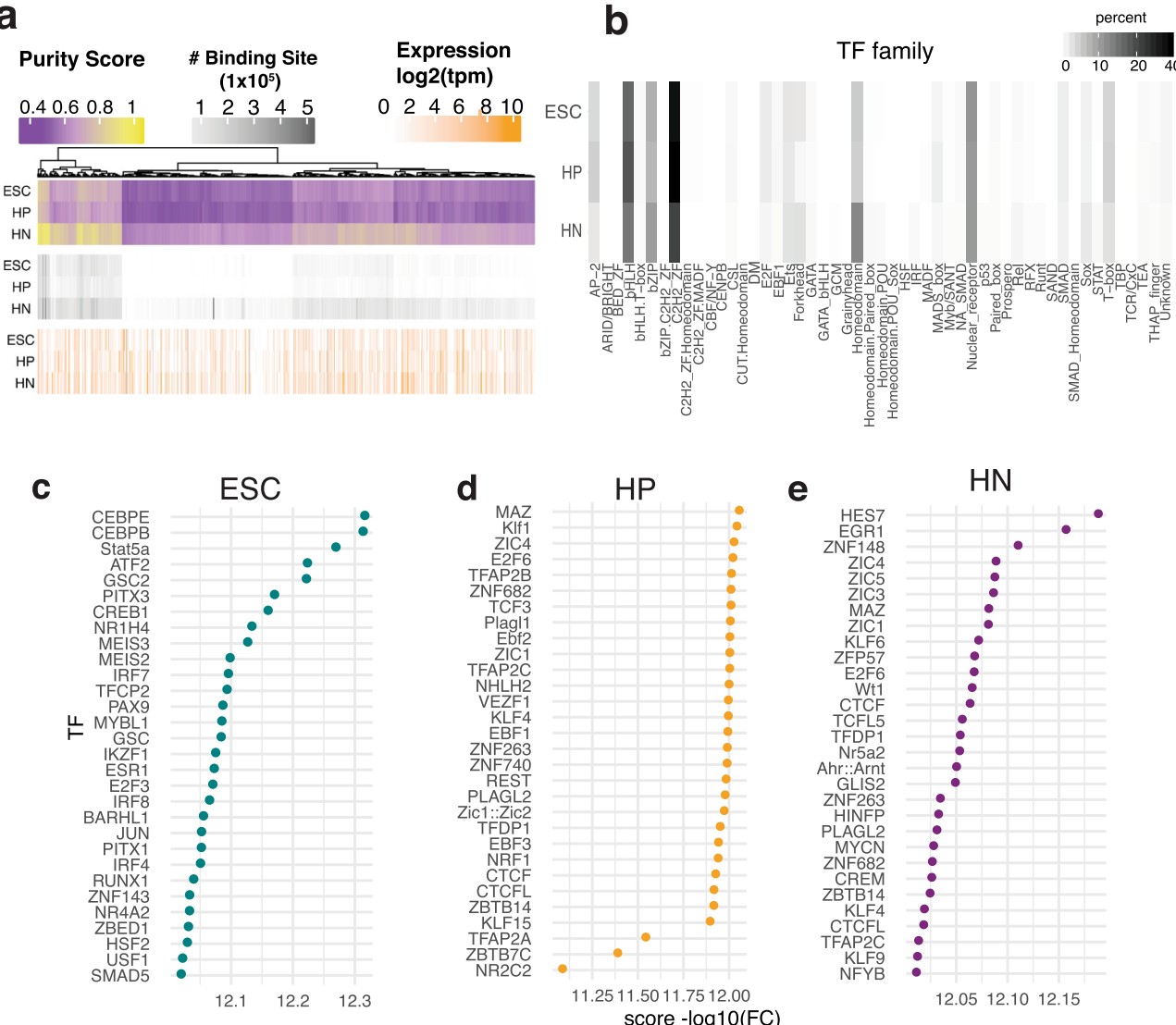

**Fig. 3 TF footprinting and analysis. a** The summary of TF-binding site prediction, the Purity scores calculated by PIQ (purple/yellow), the number of TF-binding sites passing purity cutoff >0.7 thresholds (Grayscale), log2 transformed expression of the respective TF in HN (white/orange). **b** Grouping number of TF-binding sites family. Darker color indicates a higher number of predicted binding sites. **c–e** Enrichment of TF-binding sites in putative cREs compared to other OCRs in each cell type adjusted for GC content and read count.

addition, we intersected our PIR-OCRs with the set of H3K27ac peaks found in leptin receptor-positive neurons. Approximately 29% of the H3K27ac peaks that were enriched in the leptin receptor-positive neurons overlapped with an HN cRE (Supplementary Fig. 4b). We found 634 genes expressed in HNs connected to these regions after excluding genes bait to bait interactions, including the POMC neuron-associated transcription factor *ISL1* (Supplementary Fig. 4c)[40].

**Predicting transcription factors controlling ESC-derived HN development from spatial gene regulatory architecture**. TFs regulate gene expression by binding to specific DNA sequences such as enhancers and silencers. Local chromatin accessibility is a critical determinant of where and when TFs bind to DNA[41]. To identify TFs that may bind to cREs, we leveraged PIQ, which uses chromatin accessibility profiles to improve motif score-based matching[42]. We identified putative binding sites in each stage of differentiation and observed that more binding sites were detected

in HNs compared to ESCs or HPs (Fig. 3a). After grouping TFs by family, we detected more binding sites for Homeodomain TF factors in HNs compared to ESCs or HPs (Fig. 3b). This result was expected, as neuronal identity is refined by the expression of multiple patterning genes[43]. We also observed fewer AP-1 family binding sites in HNs compared to ESCs and HPs; these TFs regulate the cell cycle in early cellular development[44].

Next, we checked for TF enrichment in cREs in each cell type to identify which TFs could mediate these promoter contacts (Fig. 2c). We compared the three stages of differentiation for enriched binding in the cREs compared to non-PIR OCRs. This approach generated a set of potentially relevant TFs involved in HN differentiation. We found 474 enriched TFs in ESCs, 122 in HPs, and 134 in HNs (Fig. 3c–e and Supplementary Data 2). While some TFs involved in DNA looping, such as MAZ and CTCF, were enriched in all three cell types, we also observed differences in expression of the top enriched TFs in each comparison such as ZBTB6, EBF2, EGR1, and ZIC/MYC family members (Supplementary Fig. 5).

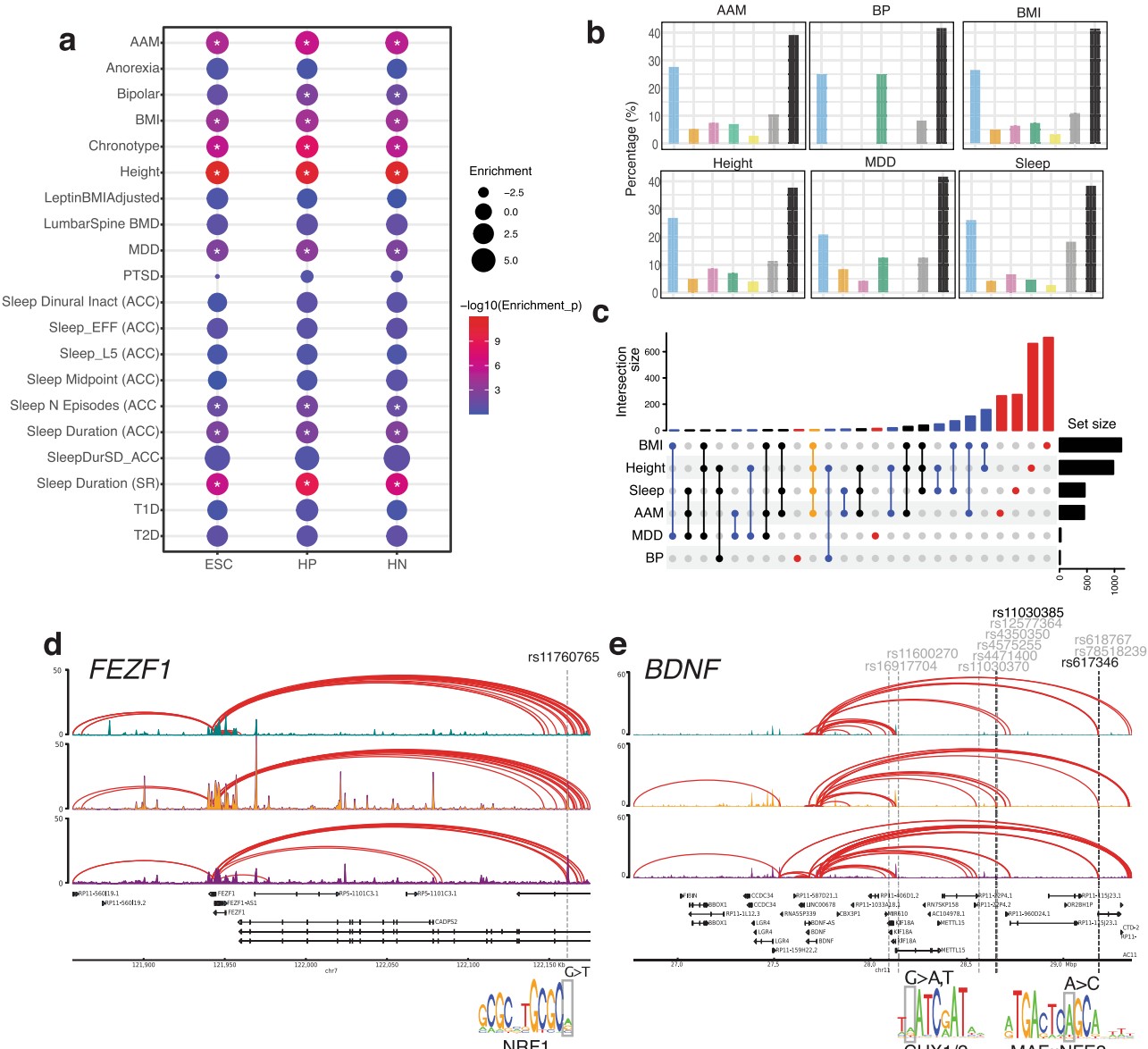

**Fig. 4 Putative *cis*-regulatory elements are enriched for GWAS signals of complex traits. a** Partitioned LD score regression of ESC, HP, and HN cRE against the indicated genome-wide signal. Circle size indicates the enrichment of estimated heritability, and color indicates statistical significance. **b** Proportion of genes from our variant-to-gene mapping located in each DE cluster (Fig. 2a) or non-DE genes (black). **c** Comparison of genes implicated in our variant-to-gene mapping analysis for each GWAS. Dots and lines indicate the intersect of the set of genes found in each GWAS. Top: the number of genes in different overlapping sets. Right: the number of SNPs detected in each GWAS. **d** *FEZF1* genomic locus with interactions connecting to a distal cRE. The SNP is located in a putative NRF1 motif. **e** Genome track for the BDNF locus. Multiple proxies in open regions are shown. Two proxies located in putative TF motifs for CUX1/2 and MAF::NFE2 are shown.

**GWAS loci enriched in ESC-derived HN cREs.** We hypothesized that cREs in our hypothalamic model are enriched for genetic variants associated with traits that are at least partly governed by the hypothalamus. We used Partitioned Linkage Disequilibrium Score Regression (LDSR)[45] to identify significantly enriched traits for associated loci falling into ESC-derived hypothalamic cREs. We assembled GWAS summary statistics from several recent studies examining metabolic, circadian, neuropsychiatric, and puberty-relevant phenotypes, and tested whether cREs were enriched for GWAS loci in at least one stage of differentiation (Fig. 4a). We detected significantly enriched signals with BMI, adult height, age at menarche (AAM), major depressive disorder (MDD), bipolar disorder, several measures of sleep (Fig. 4a and Supplementary Data 3). The

enrichment of GWAS loci for these hypothalamus-related traits in the cREs identified in the ESC-derived hypothalamic cell types highlights their utility as a model for gaining insight into the target effector genes and regulatory elements functionally related to these traits and diseases.

**Variant-to-gene mapping identifies target effector genes at GWAS-implicated loci.** Guided by the results of the partitioned LDSR analyses, we performed variant-to-gene mapping for those traits that displayed significant heritability enrichment in at least one of the three cellular differentiation stages. We began with all genome-wide significant loci in the most recent large-scale GWAS for each respective trait and queried for proxy SNPs in LD

**Table 1 Summary of variant-to-gene mapping results of GWAS signals for each trait.**

| Trait | GWAS signals | In reference panel | Unique proxies | Open proxies with *cis*-interactions[a] | Sentinels with *cis*-interactions[a] | Contacted genes[b] | Unique contacted genes[b] |
|---|---|---|---|---|---|---|---|
| Height | 3290 | 3254 | 165039 | 1081, 1098, 1073 | 594, 562, 564 | 543, 439, 507 | 981 |
| BMI | 941 | 928 | 63148 | 619, 474, 575 | 271, 211, 248 | 800, 526, 802 | 1126 |
| AAM | 499 | 463 | 13040 | 242, 194, 258 | 109, 105, 114 | 365, 267, 268 | 452 |
| Sleep | 459 | 445 | 26445 | 313, 223, 305 | 98, 84, 99 | 361, 212, 319 | 461 |
| MDD | 44 | 42 | 3059 | 22, 18, 17 | 11, 10, 8 | 15, 16, 16 | 24 |
| BP | 39 | 30 | 1697 | 1, 1, 1 | 1, 1, 1 | 2, 0, 11 | 12 |

GWAS signals: independent lead SNPs were used as sentinel SNPs. Proxies: total SNPs in LD ($R^2 > 0.6$) of lead SNPs. Open proxies with *cis*-interactions: SNPs located in cRE contacting a gene promoter. The number of sentinels with *cis*-interactions indicates the number of independent GWAS signals with an open proxy in a *cis*-interaction. The number of contacted genes per cell type, and the unique number of contacted genes from all cell types.
[a]ESCs, HPs, and HNs, respectively.
[b]Expressed at TPM > 1.

with each sentinel SNP. We overlapped this set of SNPs with the open chromatin regions identified by ATAC-seq, and queried our promoter-focused Capture C data to determine the genes in physical contact with open proxy SNPs in each of the three cell states. Finally, we filtered by expression from our RNA-seq data to limit subsequent analyses to genes expressed in at least one stage of differentiation (TPM > 1) (Table 1).

For each trait, we noticed that multiple contacted genes also have previously characterized relevant monogenic disease mutations, suggesting that our approach can identify genes with known mechanistic links to the queried traits (Supplementary Data 4). For BMI, we detected genes that are known to influence monogenic forms of extreme body weight, including *ABCC8*[46], *BDNF*[47], and *PPARG*[48]. From an AAM locus, we observed *FEZF1*, known to harbor monogenic mutations that cause delayed/absent puberty[49]. Finally, among sleep traits, our data implicated *PER2*, which encodes a factor that plays a role in advanced sleep-phase syndrome[50]. Many additional putative effector genes also have plausible biological links to each trait, while others represent novel findings in the context of these phenotypes (Supplementary Data 5).

As the cREs identified in HPs and HNs may be shared with other types of neurons, we are unable to directly identify hypothalamic-specific cREs that may be impacted by GWAS variants without chromatin accessibility and conformation data from multiple neuronal cell types. To partially address what proportion of the implicated genes may act in a hypothalamic context, we compared our results to previously published cRE maps from IPS-derived cortical neurons that were also generated in our lab from different donors[51]. Globally ~40% OCRs overlapped between HNs and iPSC-derived cortical-like neurons, with slightly more shared OCRs at promoters and less in PIR-OCRs (Supplementary Fig. 6a). We compared the expressed genes implicated by proxies located in PIR-OCRs and observed relatively little overlap for most of the six traits, with more regions associated with BMI and Height in the HN (Supplementary Fig. 6b, c). These results suggest that V2G mapping in ESC-derived hypothalamic neurons identifies distinct targets from other pluripotent stem cell-derived neurons.

**Variant contacted genes in ESC-derived HN development.** To identify variants that may impact differentiation in our model HN differentiation, we compared our contacted SNPs with the set of cREs connected to specific genes for each trait and found that ≥50% of the SNP contacted genes were differentially expressed during HN differentiation (Fig. 4b). To identify biological functions associated with these genes, we tested for GO term enrichment specific to either HP or HNs. HPs were enriched for ERK1/ERK2 cascade and phospho-inositol 3 lipid signaling

(Supplementary Data 6), which are known regulators of neural stem cell proliferation[52–54]. HNs were enriched for clathrin-dependent endocytosis and IRE1-mediated unfolded protein response (Supplementary Data 6). Endocytosis is critical for neuronal vesicle recycling at synapses and endoplasmic reticulum stress affects the response of the hypothalamus to external stimuli in obesity[55]. Thus, pathway analysis confirms that the implicated effector genes are likely to be important for hypothalamic development and function.

**Colocalization of loci associated with multiple traits.** Next, we identified contacted genes implicated in the context of multiple GWAS traits. Although most implicated genes were specific to individual traits, we identified multiple genes that were shared, suggesting a degree of overlap in the regulatory mechanisms controlling these traits (Fig. 4c and Supplementary Data 7). In particular, two loci contacted four genes (*BSN/FAM212A* and *FEZF1 /FEZF1-AS1*) which were identified in our scans of BMI, height, AAM, and sleep. To determine whether these overlaps represent likely shared regulatory regions, we performed Hypothesis Prioritization in multi-trait Colocalization (HyPrColoc) analysis for several regions. Our results highlighted both shared and distinct regulatory architectures across traits that varied by locus. For example, the *FEZF1* region colocalized among BMI, height, and AAM (regional posterior probability (PP) = 0.91), indicating a likely shared regulatory region impacting each trait (Fig. 4d). Interestingly, the proxy of the *FEZF1* signal was located in a putative NRF1-binding site, although the SNP was only predicted to have a modest effect on binding (Fig. 4d and Supplementary Data 8). In contrast to the *FEZF1* locus, although the well-known known *BDNF* was implicated as an effector gene for AAM, BMI, and sleep, these three signals appeared to be distinct (PP = 0), suggesting a complex regulatory architecture for this region that differs by trait (Fig. 4e).

**Colocalization of target effector genes with eQTLs—cumulative evidence.** Multiple data sources can contribute orthogonal evidence for effector genes at GWAS loci[56]. The GTEx consortium[31] has characterized hypothalamic tissue eQTLs, so we performed colocalization analyses to assess how many gene-SNP connections agreed with the physical variant-to-gene mapping approach in our specific cellular settings. For AAM, 13 genes colocalized with eQTLs, with two adjacent genes supported by our variant-to-gene mapping approach, *RPS26* (PP = 0.951) and *SUOX* (PP = 0.942). For BMI, we observed 12 colocalized genes, with one gene supported by our variant-to-gene mapping approach, *DHRS11* (PP = 0.822). Of the 29 genes colocalized with eQTLs for height, three were supported by our data: *NMT1*

(PP = 0.94), *RFT1* (PP = 0.85) and *RPS9* (PP = 0.75). There was only one eQTL colocalized for sleep but was not detected by our approach. For MDD and bipolar disorder, no genes colocalized with the eQTL data, which may be due to the relatively few signals detected in the eQTL analysis.

## Discussion

We used an established in vitro HN model in order to both understand its genomic architecture and to gain insight into mechanisms by which noncoding GWAS loci associated with hypothalamic-regulated traits could mediate their effects. Given the challenge in acquiring primary human hypothalamic tissue and the organ's complex makeup of cell and neuronal types[57], we leveraged RNA-seq, ATAC-seq, and promoter-focused Capture C to identify aggregation of potentially relevant *cis*-regulatory regions in ESC-derived HNs. Importantly, we verified that HNs exhibited temporal transcriptional profiles that are congruent with in vivo hypothalamic molecular expression signatures and functional networks[28].

By integrating both transcriptomics and chromatin structure at three developmental time points across hypothalamic differentiation, we defined a group of dynamic and stable promoter contacting cREs mediating gene expression changes during hypothalamic differentiation. A limitation of our study is that the HNs represent a mixed population of ARC-type neurons, so we were unable to distinguish cREs from constituent sub-nuclei. While the molecular diversity of hypothalamic neurons is beginning to be addressed in the field by single-cell transcriptomic atlases of mature and developing hypothalamus in mice[57,58] and humans[59], it is currently not feasible to generate chromatin conformation capture data on sorted hypothalamic neurons due to the relatively high number of cells required to achieve sufficient library diversity for this approach. This limitation led us to focus on identifying temporally dynamic cREs, as changes in chromatin accessibility and conformation are thought to be critical for the differentiation of a multitude of cell types[60,61]. Directly linking gene expression changes during development with cREs provides a global view of gene regulation during HN differentiation.

To identify transcriptional regulators that may bind to hypothalamic cREs, we performed TF footprinting analysis using PIQ. While we did not observe a strong correlation between predicted TF enrichment from our global analysis and score may reflect limitations of motif analysis with this analysis on heterogeneous cells, or may reflect that many of the TFs activity is regulated post-transcriptionally.

We mapped common GWAS variants associated with AAM, BMI, height, bipolar disorder, sleep, and MDD to putative effector genes via their likely cREs. This approach identified both known and novel genes. For example, *FEZF1* mutations cause hypogonadotropic hypogonadism with anosmia[49]. *FEZF1* is a zinc finger transcriptional repressor that is critical for hypothalamus development[62]. The proxy contacting the *FEZF1* promoter is located in a binding site for NRF1, a transcription factor that regulates the expression of several genes involved in mitochondrial biosynthesis and respiration, but is also important for neuronal differentiation and axogenesis[63]. *FEZF1* mutations impair puberty by disrupting the migration of gonadotropin-releasing hormone neurons, which are necessary to initiate puberty, from the olfactory bulb placode to the hypothalamus during fetal development[49]. In contrast to *FEZF1*, while *BDNF* was implicated in three traits, we observed distinct GWAS association landscapes, with different sentinels pointing to different proxies that consistently contacted the *BDNF* promoter. Thus, *BDNF* appears to have an intricate regulatory architecture and harbors

multiple trait-associated variants that likely act in cell-type and temporally specific contexts. Finally, among sleep traits, our data implicated *PER2*, which encodes a factor that plays a role in advanced sleep-phase syndrome[50].

To uncover genes implicated by multiple analytic approaches, we also performed colocalization analyses of the implicated traits with hypothalamic eQTLs. Both eQTL and variant-to-gene mapping approaches identified *DHRS11* for BMI. The overlap between the two approaches was low, possibly due to differences between ex vivo tissue samples and stem cell-derived cells. Methods like eQTL analyses and chromatin conformation capture often map genetic variants to multiple candidate effector genes. While eQTL associates effector genes by associating genotype and gene expression, it commonly suffers from low statistical power. On the other hand, the chromatin conformation map only demonstrates physical contact but fails to indicate the regulatory consequence of specific allele on gene expression. As such these putative connections represent a first step in uncovering the effector genes at GWAS loci, and warrant further functional follow-up. A confluence of evidence is critical for distinguishing true effector genes from the many "bystander genes" identified in eQTL studies;[56] our physical variant-to-gene mapping pipeline represents one such approach.

In addition to the molecular heterogeneity of the hypothalamus, there are several limitations to our study. While the hypothalamus is known to exhibit sex-specific differences in cell composition and activity, our cells were derived from the female H09 ESC cell line prevents this analysis. Another limitation in our experimental design is that we only examined neurons generated to resemble a single brain region, which limits our ability to distinguish cREs that may be shared across cell types or specific to a hypothalamic context. Due to this limitation, it is likely that some GWAS associations intersecting HP/HN cREs may be common to neural progenitors/young neurons from multiple brain regions or not represented in vivo hypothalamic neurons.

In addition, HNs do not directly correspond to fully mature neurons found in the adult hypothalamus and display expression of markers associated with prenatal mouse neurons[28]. Reaching advanced stages of differentiation remains a challenge in both iPSC and organoid models[64]; however, these HNs are functionally active and respond to hormones such as leptin and insulin, and thus represent an accessible human hypothalamus model[28]. As a result of the limitation on neuronal maturity, some of our results are likely specifically relevant to prenatal neurons. Exposure to maternal obesity or gestational diabetes is associated with future weight gain via alternations to the hypothalamus[65], suggesting that different stages of hypothalamic development might be particularly relevant in the context of BMI. Further improvements to neuronal differentiation and organoid protocols may allow later stages of differentiation to be reached, which would facilitate comparisons between young and more mature neurons.

Here, we report aspects of the genomic architecture of a stem cell-based model of human hypothalamic development. We relate this architecture to the cellular ontogenesis of the human hypothalamus, and to the regulation of genes that influence complex phenotypes. Application of these strategies enables specific gene attributions for noncoding SNPs implicated in relevant common traits by GWAS efforts. These integrated datasets, therefore, offer valuable insight for prioritizing candidate genes that drive the molecular mechanisms by which the hypothalamus contributes to the pathogenesis of relevant complex traits.

## Methods

**Human ESC-derived hypothalamic neuron differentiation**. The HN differentiation protocol was described previously[28]. Briefly, the human ESC H9 line was seeded on Matrigel plates (16 million cells/148 cm²; 5 × 148 cm² Corning dishes) in

ESC medium (KnockOut DMEM supplemented with 15% knockout serum replacement, 0.1 mM MEM non-essential amino acids, 2 mM GlutaMAX, 0.06 mM 2-mercaptoethanol) with FGF-basic (AA 1–155), (20 ng/ml media) and 10 μM Y-27632. Upon confluency (day 1), cells were cultured in ESC medium without FGF-basic and Y-27632, but supplemented with Shh (100 ng/ml), purmorphamine (2 μM), 10 μM SB431542, and 2.5 μM LDN193289. From days 5 to 8, ESC medium was gradually replaced with neuroprogenitor medium (DMEM/F-12 supplemented with 0.1 mM MEM non-essential amino acids, N-2 Supplement, 0.2 μM ascorbic acid, 0.16% glucose). On day 9, cells were switched into neuronal differentiation medium (DMEM/F-12 supplemented with 0.1 mM MEM non-essential amino acids, N-2 supplement, B-27 supplement minus vitamin A, 0.2 μM ascorbic acid, 0.16% glucose containing 10 μM DAPT). On day 12, cells were collected with TrypLE Express Enzyme at 37 °C for 7 min and washed twice including filtration through pre-wetted 40-μm Corning sterile cell strainer. The hypothalamic progenitor cell pellet was then resuspended with neuronal differentiation medium containing 10 μM Y-27632 for plating on 148-cm$^2$ dishes coated with poly-L-ornithine solution (0.01%) and laminin (4 μg/ml) at a seeding density of 16 million cells/148 cm$^2$. After 4 h, the medium was changed to neuronal differentiation medium supplemented with 10 μM DAPT. On day 15, the neuronal differentiation medium was supplemented with 20 ng/ml BDNF until collection on day 27.

**Immunocytochemistry and imaging of human ESC-derived hypothalamic neurons.** The human ESC H9 line was differentiated using the protocol above, the only distinction being that they were re-plated on day 12 into 24-well plates (Thermo Scientific Nunc) at a seeding density of 200,000 cells per well.

Differentiated hypothalamic neurons were fixed in 4% paraformaldehyde, PBS for 20 min at room temperature (RT), followed by two washes with PBS. They were incubated (to permeabilize and block) for 1 h at RT with buffer containing 10% normal donkey serum, 0.1% Triton X-100, PBS. Primary antibodies (goat polyclonal to POMC, ab32893, 1:200; rabbit polyclonal to tubulin beta 3 (TUBB3), Biolegend 802001, 1:1000) were diluted in this buffer. Cells were incubated with primary antibody solution overnight at 4 °C. After two washes with 0.1% Triton X-100, PBS incubation with secondary antibodies (anti-rabbit 488 Alexa at 1:1000, anti-goat 555 Alexa at 1:1000) and the nuclear marker Hoechst (1:5000) was performed in PBS for 2 h at RT. After two washes with PBS, cells were stored in PBS at 4 °C until imaging.

Images were taken using an Olympus IX73 inverted microscope (×40 objective).

**ATAC-seq, RNA-seq, Capture C library generation, processing, peak calling**
*ATAC-seq library generation.* A total of 50,000 cells were centrifuged at 550 × g for 5 min at 4 °C. The cell pellet was washed with cold PBS and resuspended in 50 μL cold lysis buffer (10 mM Tris-HCl, pH 7.4, 10 mM NaCl, 3 mM MgCl$_2$, 0.1% NP-40/IGEPAL CA-630) and immediately centrifuged at 550 × g for 10 min at 4 °C. Nuclei were resuspended in the Nextera transposition reaction mix (25 μL 2× TD Buffer, 2.5 μL Nextera Tn5 transposase, and 22.5 μL nuclease-free H2O) on ice, then incubated for 45 min at 37 °C. The tagmented DNA was then purified using the Qiagen MinElute kit and eluted in 10.5 μL elution buffer (EB). Ten microliters of purified tagmented DNA were PCR amplified using the Nextera Indexing Kit for 12 cycles to generate each library. The PCR reaction was subsequently purified using 1.8x AMPure XP beads, and concentrations were measured by Qubit Fluorometer. The quality of completed libraries was assessed on a Bioanalyzer 2100 high sensitivity DNA Chip. Libraries were paired-end sequenced at the Center for Spatial and Functional Genomics on the Novaseq 6000 platform (51 bp read length).

*ATAC-seq analysis and peak calling.* The number reads from the hypothalamic neurons were downsampled to make the sequencing depth comparable between conditions using sambamba. Open chromatin regions were called using the ENCODE ATAC-seq pipeline. Pair-end reads from all replicates for each cell type were aligned to the hg19 genome using bowtie2, and duplicate reads were removed from the alignment. Aligned tags were generated by modifying the reads alignment by offsetting +4 bp for all the reads aligned to the forward strand, and −5 bp for all the reads aligned to the reverse strand. Narrow peaks were called independently for pooled replicates for each cell type using macs2 (-p 0.01 --nomodel --shift -75 --extsize 150 -B --SPMR --keep-dup all --call-summits) and ENCODE blacklist regions were removed from called peaks. We then merged peaks with at least 1 bp overlap between replicates to generate a consensus set of peaks. The consensus set peaks were filtered to those which were reproducible in at least half the ATAC-seq replicates using bedtools intersect[10]. For analyses involving cell-type-specific sets of peaks, we considered the set of consensus peaks with mean FPKM value greater than 1 to be "open" in that cell type.

For TF analysis replicated, de-duplicated ATAC-seq bam files were merged and downsampled to consistent read count for each stage of differentiation to calculate purity scores for each TF.

*Differential analysis of chromatin accessibility.* To identify differentially accessible OCRs between ESCs, HPs, and HNs, we used the R package csaw, which uses the de-duplicated read counts for the consensus OCRs for each replicate to normalized against background (10 K bins of the genome). OCRs with the median value of less

than 1.2 CPM (~10–50 reads per OCR) across all replicates were removed from the further differential analysis. Similar to RNA-seq differential analysis, accessibility differential analysis of the consensus OCRs was performed using glmQLFit approach, fitting cell type in edgeR, but using the csaw normalization scaling factors. Differential OCRs between cell types were identified with thresholds of FDR < 0.05 and absolute log2 fold change >1. FPKM values were calculated for all OCRs in the consensus list.

**RNA-seq library generation and analysis**
*RNA-seq library generation.* RNA was isolated from each cell type in triplicate using TRIzol Reagent. RNA was then purified using the Direct-zol RNA Miniprep Kit and depleted of contaminating genomic DNA using DNAse I. Purified RNA was then checked for quality on the Bioanalyzer 2100 using the Nano RNA Chip, and samples with a RIN number above 7 were used for RNA-seq library synthesis. RNA samples were depleted of rRNA using the QIAseq FastSelect RNA Removal Kit then processed into libraries using the NEBNext Ultra II Directional RNA Library Prep Kit for Illumina according to the manufacturer's instructions. The quality and quantity of the libraries were measured using the Bioanalyzer 2100 DNA chip and Qubit Fluorometer. Completed libraries were pooled and sequenced on the NovaSeq 6000 platform using paired-end 51 bp reads at the Center for Spatial and Functional Genomics at CHOP.

*RNA-seq processing and differential expression analysis.* Sequencing data were demultiplexed and FastQ files were generated using Illumina bcl2fastq2 conversion. Paired-end Fastq files for each replicate were mapped to the reference genome using STAR. Gene features were assigned to a curated annotation consisting of GencodeV19 with lincRNA and sno/miRNA annotation from the UCSC Table Browser. The raw read count for each gene feature was calculated using HTSeq-count. with parameter settings -f bam -r pos -s reverse -t exon -m intersect. The genes located on chrM or annotated as ribosomal RNAs were removed before further processing.

Differential analysis was performed in R using the edgeR package. Briefly, the raw reads per genes features were converted to read Counts Per Million mapped reads (CPM). The gene features with the median value of less than 0.7 CPM (10–18 reads per gene feature) across all samples were filtered. Normalization scaling factors were calculated using the trimmed mean of the M-values method. Differentially expressed genes between ESCs, HPs, and HNs were identified with thresholds of FDR < 0.05 and absolute log$_2$FC > 1. Expression values are reported as transcript per million mapped reads (TPM). We clustered standardized TPM values of differentially expressed genes using the R function hclust. Genes expression values were standardized using the R function, scale. Following this, the top six branches were cut to define the clusters used in subsequent comparisons.

**Capture C library generation and analysis**
*3C library generation.* We used standard methods for 3C library generation[66]. For each library, 10$^7$ fixed cells were thawed at 37 °C, followed by centrifugation at RT for 5 min at 1845 × g. The cell pellet was resuspended in 1 mL of dH$_2$O supplemented with 5 μL 200× protease inhibitor cocktail, incubated on ice for 10 min, then centrifuged. The cell pellet was resuspended to a total volume of 650 μL in dH2O. In total, 50 μL of cell suspension was set aside for pre-digestion QC, and the remaining sample was divided into three tubes. Both pre-digestion controls and samples underwent a pre-digestion incubation with the addition of 0.3% SDS, 1× NEBuffer DpnII, and dH$_2$O for 1 h at 37 °C in a Thermomixer shaking at 1000 rpm. A 1.7% solution of Triton X-100 was added to each tube, and shaking was continued for another hour. After the pre-digestion incubation, 10 μL of DpnII was added only to each sample tube, and continued shaking along with the pre-digestion control until the end of the day. An additional 10 μL of DpnII was added to each digestion reaction and digestion continued overnight. The next day, another additional 10 μL of DpnII was added and the incubation continued for another 2–3 h. In total,100 μL of each digestion reaction was then removed, pooled into one 1.5-mL tube, and set aside for digestion efficiency QC. The remaining samples were heat-inactivated at 65 °C for 20 min at 1000 rpm in a Thermomixer and cooled on ice for 20 additional minutes. Digested samples were ligated with 8 μL of T4 DNA ligase and 1× ligase buffer at 1000 rpm overnight at 16 °C in a Thermomixer. The next day, an additional 2 μL of T4 DNA ligase was spiked into each sample and incubated for another few hours. The ligated samples were then de-crosslinked overnight at 65 °C with Proteinase K along with the pre-digestion and digestion controls. The following morning, both controls and ligated samples were incubated for 30 min at 37 °C with RNase A, followed by phenol/chloroform/isoamyl alcohol (Fisher Cat # BP1752I400) extraction and ethanol precipitation at −20 °C. The 3C libraries were centrifuged at 1000×g for 45 min at 4 °C, while the controls were centrifuged at 1845×g, to pellet the samples. DNA pellets were resuspended in 70% ethanol and again centrifuged as described above. The 3C library pellets and control pellets were resuspended in 300 μL and 20 μL dH$_2$O, respectively, and stored at −20 °C. Sample concentrations were measured by Qubit Fluorometer. Digestion and ligation efficiencies were assessed by gel electrophoresis on a 0.9% agarose gel and quantitative PCR (Brilliant III SYBR qPCR Master Mix, VWR Cat # 97066-528).

*Promoter capture library generation.* We followed our same protocols as previously published[20]. Isolated DNA from 3C libraries was quantified using a Qubit Fluorometer, and 10 µg of each library was sheared in dH$_2$O using a QSonica Q800R to an average fragment size of 350 bp. QSonica settings used were 60% amplitude, 30 s on, 30 s off, 2 min intervals, for a total of five intervals at 4 ℃. After shearing, DNA was purified using AMPure XP beads. DNA size was assessed on a Bioanalyzer 2100 using a DNA 1000 Chip and DNA concentration was checked via Qubit Fluorometer. SureSelect XT library prep kits were used to repair DNA ends and for adaptor ligation following the manufacturer's protocol. Excess adaptors were removed using AMPure XP beads. Size and concentration were checked again by Bioanalyzer 2100 using a DNA 1000 Chip and by Qubit Fluorometer before hybridization. One microgram of the adaptor-ligated library was used as input for the SureSelect XT capture kit using manufacturer protocol and our custom-designed 41 K promoter Capture-C probe set. The quantity and quality of the captured libraries were assessed by Bioanalyzer using a high sensitivity DNA Chip and by Qubit Fluorometer. SureSelect XT libraries were then paired-end sequenced on Illumina NovaSeq 6000 platform (51 bp read length) at the Center for Spatial and Functional Genomics at CHOP.

*Analysis of Capture C.* Paired-end reads from three replicates from ESCs, HPs, and HNs were pre-processed using the HICUP pipeline with the default parameters. Reads were aligned to hg19 using bowtie2. We called call significant promoter interactions using the read count from promoters included in our reference bait. As previously reported[20], significant interactions were called using CHiCAGO with default parameters except for bin-size set to 2500. In addition to our analysis per individual DpnII fragment (1frag), we also called interactions by binning four fragments, which improves detection of long-distance interactions. Significant interactions at 4-DpnII fragment resolution were also called using CHiCAGO. Interactions with a CHiCAGO score >5 in at least one cell type in either 1-fragment or 4-fragment resolution were considered significant.

**Quality control metrics.** Reproducibility between ATAC-seq and RNA-seq samples was determined by principal component analysis and pairwise Pearson correlation coefficients between samples. Median expression values were downloaded from GTEx (v7). The spearman rank correlation of the genes with expression level pass a threshold of TPM > 5 in at least one cell/tissue type (16,953 genes) were calculated using Spearman's Correlation Coefficient (cor function in R). Cell-specific enrichment analysis was conducted using the CSEA tool webserver (http://genetics.wustl.edu/jdlab/csea-tool-2/)[33]. For ATAC-seq fragment distribution plots were examined for the presence of mono-nucleosome and di-nucleosome peaks to verify successful Tn5 transposition.

Genomic annotations: Promoters were defined as 1500 kb upstream and 500 kb downstream of the TSS (Genecode V19). Overlapping annotations were assigned to genomic features based on a hierarchy of (1) Promoter, (2) 5'UTR, (3) CDS, (4) 3'UTR, (5) first intron, (6) other introns, or (7) intergenic. The percentage of OCRs overlapping with each feature was visualized as pie charts using ggplot2. All coordinates refer to hg19 as the reference genome. Genome tracks were visualized using the python package pyGenomeTracks version 3.0.

**Variant-to-gene mapping.** Sentinel SNPs were collected from the most recent large-scale GWAS studies. Proxies for each sentinel were queried using SNiPa using the following parameters: Genome assembly GRCh37; Variant set 1000 Genomes, Phase 3 v5; Population European; Genome annotation Ensembl 87 and $r^2 > 0.6$. Intersection as done previously[20]. We identified proxies located in open chromatin and fragments interacting with a bait using bedtools intersect. We considered all interactions with a proxy SNP located in a distal interaction fragment and those falling within OCRs located in baits. Putative target effector genes were then filtered by expression in each respective cell state (TPM > 1). The same parameters were used for variant-to-gene mapping hypothalamic traits with previously published IPS-derived neuron dataset[51]. These genes were functionally annotated by the DAVID functional annotation tool.

**Gene set enrichment.** GO and REACTOME datasets annotated in MSigDB (v7.0) were used for gene set enrichment analyses. Statistical significance of gene set enrichment was determined using the hypergeometric test, implemented in the R phyper function.

**Transcription factor analysis.** PIQ, which integrates TF motif scanning with TF footprinting using DNAase or ATAC-seq data, was used to predict TF-binding sites[42]. We scanned JASPAR2020 core[67] PWMs against hg19, with ENCODE blacklist regions excluded using the default settings. For downstream analyses we considered TF-binding sites passing the default cutoff of purity >0.7. We identified TF motifs enriched in cREs compared the set of non-PIR-OCRs using the R package BiFET (v 1.4.0), with a cutoff of FDR < 0.05,

**Partitioned LD score regression.** Partitioned heritability was measured using LD Score Regression v1.0.0[45]. Partitioned LDSR requires the GWAS summary statistics and a feature annotation. ESC, HP, and HN annotations were generated using bed files containing positions of the cRE (promoter OCRs + PIR-OCRs) with + /−500 bp extension as previously performed[45]. We selected a set of traits related to metabolic, endocrine, and neuropsychiatric traits with available GWAS summary statistics (see Supplementary Methods).

**Comparison with mouse hypothalamic epigenetic data.** We retrieved the processed data from GEO (accession GSE112125). We used liftover to convert mm9 coordinates to hg19 with the similarity cutoff -minMatch=0.1. We excluded the top 1% longest peaks for both H3K27ac ChIP-seq and ATAC-seq data. We used the R package regioneR (version regioneR 1.22.0). To perform permutation tests to determine if the accessible regions with H3K27ac+ were enriched in the dataset (10,000), we tested for overlap between the set of HN cREs and set of H3K27ac+ peaks that were significantly enriched in LepR+ neurons across conditions (FDR < 0.05).

**GWAS colocalization.** Summary statistics for six regions with overlapping associations for 3–4 input traits were imputed using FIZI. Common variants (MAF ≥ 0.01) from the European ancestry 1000 Genomes Project v3 samples were used as a reference panel for the imputation. Default parameters were used with the exception that the minimum proportion parameter was lowered to 0.01. Standard errors and betas for the imputed SNPs were estimated using the method from https://github.com/zkutalik/ssimp_software/blob/master/extra/transform_z_to_b.R. Subsequently, HyPrColoc was used to test for colocalization across all input traits simultaneously. Separately, we tested for colocalization for each input trait genome-wide against GTEx v.7 hypothalamic eQTLs using coloc[68].

**Reporting summary.** Further information on research design is available in the Nature Research Reporting Summary linked to this article.

## Data availability

Further information and requests for reagents should be directed to and will be fulfilled by the lead contacts, Struan F.A. Grant and Diana L. Cousminer. All reagents and software used are listed in Supplementary Data 9. The raw and processed ATAC-seq, Capture C, and RNA-seq data described in this study are deposited in the gene expression omnibus (GEO) with the accession number GSE152098. Public datasets accessed and used in the study: JASPAR2020: http://jaspar.genereg.net/downloads/; GTEX v7: https://gtexportal.org/home/datasets; Mouse Sorted Hypothalamic ATAC-seq and H3K27ac Chip-seq datasets: https://www.ncbi.nlm.nih.gov/geo/query/acc.cgi?acc=GSE112125; LD reference panels: https://github.com/bulik/ldsc; Molecular Signatures Database (MSigDB) v7: https://www.gsea-msigdb.org/gsea/msigdb/index.jsp. We accessed publicly available GWAS summary stats: age at Menarche: https://www.reprogen.org/data_download.html; anorexia: https://www.med.unc.edu/pgc/download-results/; bipolar disorder: https://www.med.unc.edu/pgc/download-results/; body mass index: https://portals.broadinstitute.org/collaboration/giant/index.php/GIANT_consortium_data_files; chronotype: http://www.t2diabetesgenes.org/data/; height: https://portals.broadinstitute.org/collaboration/giant/index.php/GIANT_consortium_data_files; major depressive disorder: https://www.med.unc.edu/pgc/download-results/; post-traumatic stress disorder: https://www.med.unc.edu/pgc/download-results/; pubertal growth: https://egg-consortium.org/; self-reported sleep: http://kp4cd.org/datasets/sleep; accelerometer-associated sleep traits: http://www.t2diabetesgenes.org/data/; type II diabetes: https://cnsgenomics.com/content/data.

## Code availability

Publicly available analysis software and code were used as described in "Methods".

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

## Acknowledgements

We acknowledge Elisabetta Manduchi for establishing the Capture C pipeline. The project described was supported by the National Center for Research Resources, Grant UL1RR024134, and is now at the National Center for Advancing Translational Sciences, Grant UL1TR000003. The content is solely the responsibility of the authors and does not necessarily represent the official views of the NIH. Supported in part by the Institute for Translational Medicine and Therapeutics' (ITMAT) Transdisciplinary Program in Translational Medicine and Therapeutics. D.L.C. is supported by the NICHD (NIH1K99HD099330-01). R.L.L. is supported by DK52431-23 and P30DK026687-41. S.F.A.G. is supported by R01 HD056465, R01 HG010067, R01 HL143790, and the Daniel B. Burke Endowed Chair for Diabetes Research.

## Author contributions

M.C.P. processed sequencing data and conducted bioinformatic analyses of functional genomic data. D.L.C. performed variant-to-gene mapping and GWAS colocalization analyses. K.M.H. collected all relevant material and generated ATAC-seq libraries. S.H.L. contributed to data analysis and appraisal. M.E.L. generated 3C libraries and performed Capture C. S.L. generated RNA-seq libraries. J.A.P. sequenced the samples and contributed to generating 3C and ATAC-seq libraries. J.P.B. contributed to colocalization analyses. M.C.D.R. and A.B. contributed to generating differentiated cells, immunocytochemistry, and imaging. K.B., C.L., and M.E.J. contributed to lab processes. C.S. and A.C. contributed to the pipeline for processing sequencing data. R.K.H. contributed to pathway analyses. C.A.D., R.R., and R.R.L. generated the differentiated cells used in sequencing efforts. S.H.L., R.I.B., A.D.W., B.F.V., and R.R.L. provided critical feedback. M.C.P., C.A.D., S.H.L., K.H.M., D.L.C., and S.F.A.G. conceived the project and wrote the paper with input from all authors.

## Competing interests

The authors declare no competing interests.
