## [Peer Review File · Nature Communications]

Title: Cis-regulatory architecture of human ESC-derived hypothalamic neuron differentiation aids in variant-to-gene mapping of relevant complex traitsREVIEWER COMMENTS

Reviewer #1 (Remarks to the Author):

In this study, the authors generate physiologically relevant genomic datasets for complex traits studied by GWAS, the authors use a hypothalamic neuron differentiation protocol to generate corresponding RNA-seq, ATAC-seq, and Promoter Capture-C datasets that they then analyze and correlate to disease-associated SNPs and eQTLs from published datasets.

A general analysis of the promoter-interacting cREs in ESC/HP/HN cells shows some general enrichment for some TFBS, but nothing striking. However, by obtaining disease-relevant genomic data the authors improve the interpretation of the GWAS hits reported by previous studies and can link them to genes that have been shown to be disease-relevant in functional studies.

The manuscript is extremely well written, and in contrast to many other genomic papers, the logical links of the analysis are easy to follow and sound. Moreover, the authors clearly state results and their relevance but do not make attempts to oversell their data. Such genomic datasets from disease-relevant cell types are important to harvest the information obtained from large GWAS-studies and provide an important resource for further functional studies.

Before publication, I suggest following minor changes to the text and figures to improve the readability of the study:

Minor points:

1. Despite having a considerable amount of whitespace, the authors avoid labeling individual tracks and displays in the Figures, but rather present all the information in the Figure legends. I strongly suggest adding some more information to the individual panel labels, as it is easier to understand for the reader. E.g. adding RNA-seq and ATAC-seq to the graphs in Fig. 1B/C and to the PCAs in Fig. 1a/b, adding differential accessibility or something similar to Fig. 2D. There could be more similar changes.
2. Figure 3a and Text: It seems as if the expression level of the respective TFs does not correlate so well with the Purity score or number of binding sites for the respective TF. This appears to be a limitation of the bioinformatic tool or purity of their cell populations. The authors should mention this.
3. Table 1: Explicitly clarify the column labeling in the manuscript text or rename: what are proxies? What are open proxies?
4. The authors should include an overview Supplemental Figure representing the interaction profile observed in the 3 different cell types. How many promoter-interacting regions did they find in each cell type and what was the mean number of interacting OCRs per gene?
5. Although the heterogeneity in especially HN neurons is addressed in the discussion section, this could also be mentioned at the respective results section. This will help the reader to evaluate/interpret the presented data.

Daniel Ibrahim

Reviewer #3 (Remarks to the Author):

In their manuscript, Pahl and Doege et al. apply RNA-seq, ATAC-seq, and Capture C to ESC-derived hypothalamic-like precursors (HPs) and neurons (HNs) to generate a map of chromatin accessibility in the pluripotent stage and at these two stages. They first examine the expression of several transcription factors known to have roles in hypothalamic development, such as NKX2.1, within in their model system. After validating their model, they proceed to identify both genes and open chromatin regions that are differentially regulated across these developmental stages, allowing the discovery of putative cis-regulatory elements (cRE) that interact with differentially expressed genes. By considering chromatin accessibility, the authors identify transcription factors that bind in these putative cREs to characterise the developmental transcription-regulation landscape. They conclude by showing that some putative cREs are co-localised with GWAS hits linked to disorders involving the hypothalamus. While the approach of integrating several data forms into a single, coherent dataset is conceptually interesting, and the authors should be commended on assembling a clearly written manuscript with a nicely detailed materials and methods section. However, the study remains preliminary in its current state and would be improved if several major issues were addressed.

Major comments

1. The authors focus on human pluripotent stem cell (hPSC)-derived hypothalamic progenitors and young neurons as a model system. This is undeniably an interesting region of the brain to examine, but it is unclear which of their results might have been obtained by comparing hESCs to generic neural progenitors and generic young neurons, rather than hypothalamic progenitors and hypothalamic neurons specifically. Indeed, the author's comparison to GTEX reveals a strong correlation of their cells to not just to hypothalamus but also to other brain regions (Fig. 1E). In order to make conclusions about cREs relevant to hypothalamic cell function, the authors should perform similar analysis on hPSCs differentiated to other neuronal lineages (e.g. cortical, midbrain, spinal cord) for which robust differentiation protocols exist. Indeed, this comparison might give them greater power to reveal which cREs are hypothalamus-specific which would aid in the interpretation of GWAS coloc analysis for traits thought to be tied to the function of hypothalamic cells
2. The authors identify only a modest number of colocalizations and appear to have selected GWAS based on the likely involvement of hypothalamic cells (e.g. growth/height, BMI, and sleep, although bipolar disorder, PTSD, and depression are also included) and are a mixture of well-powered studies, and less well-powered studies. In order to better interpret whether the author's data provides insights into the specific traits they wish to examine, it would be interesting to compare to other GWAS relevant to the nervous system but not hypothalamus (e.g. schizophrenia) and ask if there is any enrichment in colocs for hypothalamus-associated traits. If the authors they had data from hPSCs differentiated to other brain regions (see comment 1), this comparison would be even more valuable.
3. One potentially useful application of the integrated dataset would be to leverage ATAC-seq and chromatin conformation capture data to expand the list of genes nominated by GWAS based on long-range interactions. For example, are there any examples of genes that might be affected by at a distance from GWAS loci and that have coding variants of large effect in traits relevant to the hypothalamus (e.g. height, BMI) in human datasets?

4. The authors focus on bulk methods for data collection yet the hypothalamus is developmentally complex composed of a multitude of individually rare cell populations. While it is true that single-cell methods for chromatin capture are limited, single-cell RNAseq and ATAC-seq are well-established and could provide additional value to their study to help interpret whether any co-localisation might be cell type dependent, and to measure heterogeneity in their samples to estimate which cell populations are contributing most to the described cREs. Furthermore, it's not clear why the specific time points in differentiation were selected and how closely these cell populations reflect those found in the adult. For example, how does the maturity of cells produced in this study compare to hypothalamic developmental datasets from the mouse (e.g. Kim et al, Nature Communications, 2020)? In my opinion, it's difficult to ascribe likely biological relevance to the results since the study has no comparison groups and is sampling a complex mix of cells whose relevance to the adult human hypothalamus is not clearly established.
5. The authors appear to have performed the study using three distinct differentiations of a single cell line (H9 or WA09) at only two time points in the differentiation process. Given the well-known variability in differentiation propensity between cell lines, biological replicates should be generated using at least 3 distinct genetic backgrounds. Otherwise, conclusions should be drawn more narrowly and discussed in the context of the WA09 cell line at the specific differentiation time points tested.
6. The authors demonstrate modest success of confirming that some known GWAS hits are co-localised with cREs, but the reader is left unsure which of the results are sufficiently compelling to warrant future studies. Any further validation of their by an orthogonal method or demonstration of relevance to the co-localised traits would greatly increase the potential impact of the study.

Minor comments

1. The term cRE is not defined at first mention (line 140) and is sometimes inconsistently called cisRE (line 802, Fig S3)
2. The discussion of FEZF1 and its association with Kallman syndrome seems like a stretch, since any cRE are likely to modestly affect FEZF1 expression and Kallman syndrome is an autosomal recessive disease requiring severe impairment of gene function.
3. In Figure 1D, it is difficult to see that there is a difference in the number of interaction arcs, and which of these changes are significant. Also, the colour scheme of HP and HN in 1D is inconsistent with that in 1A and 3D,E.
4. In Figure 2C, it is difficult to distinguish between those peaks that are dashed (for PIR-OCR) and those that are not (for non PIR-OCR). Additionally, Figure 2D could be made more clear by the addition of cell type transition labels, similar to those in Figure 2E.
5. In the legend for Figure 4, the description for panels B and C appears to have been swapped.

Reviewer #4 (Remarks to the Author):

In this study the authors analyzed cis-regulatory architecture in an in vitro model of hypothalamus embryonic and fetal development. The authors used a human embryonic stem cell line and applied a

published protocol for hypothalamic neuron differentiation [1]. This generated a hypothalamic progenitor (HP) cell sample and a hypothalamic neuron (HN) cell sample. These samples were studied using RNA-seq, ATAC-seq and Capture C. The identified loci were cross-referenced with published GWAS SNPs from studies on phenotypes controlled by hypothalamus function, for example body weight, height, and onset of puberty.

The authors were able to find genes that have a known function in these phenotypes. This shows the utility of in vitro models - in the absence of primary tissues - to study regulatory networks. It also shows the strength of using several levels of genomic information, such as gene expression, chromatin openness and conformation, as well as SNP associations. While the in vitro models have inherent shortcomings and represent a simplified model of neuronal cell, they have the advantage of being more amenable to molecular studies.

The major strength of this paper lies in showing that in vitro models do have some usefulness. As the authors show, significant signals can be detected that can be connected to the hypothalamus and its function. The strength lies in the availability of starting material, which allowed execution of these analyses. Another strength of the paper is the combination of different layers of information, for example cross-referencing cis-regulatory regions with GWAS signals. Studies of this kind are very timely and useful to the field of genetics of metabolism and endocrine signaling.

A few changes could improve the manuscript even further. For example, Inoue et al. published a study in 2019 studying the epigenome of leptin-responsive genes in the mouse hypothalamus [2]. While this study did not look at human tissue, but mouse tissue, it would still be interesting to compare the identified loci, since it is the only study to my knowledge that also looked at the epigenetics and not only gene expression of the hypothalamus. If there is an overlap between loci, this would strengthen the findings further. It would also be interesting to see if the same loci are implicated in development (present manuscript) and young adult stage (Inoue).

Other published studies that could be referenced in the manuscript are Willer et al., Nat Genet 2009; Ong et al., Nat Genet 2009; and Cousminer et al., Hum Mol Genet 2013 [3,4,5]. These would fit well in line 60-61 when the authors cite other GWA studies. The currently referenced studies center around sleep and chronotype. I believe it would be fitting to also reference GWA studies related to other phenotypes regulated by hypothalamus, such as body weight and puberty.

Especially since the authors study the hypothalamus, which as they say controls features such as reproductive timing and body weight. Those features show sex differences, which raises the question of whether the experimental design should have included one XX and one XY cell line. While this cannot be changed now without repeating the entire study, it is worth considering whether to make it clearer for the reader in the manuscript that results come from one XX cell line and its potential limitations.

Also, with regards to the cellular samples that are the input material into the genomics assays and the basis of all analyses, more information about the starting material would strengthen the manuscript. Supp Fig 1E shows RNA expression of some marker genes as quality control, but since they are basically studying a cell model, it would be nice to see a figure with images of the cells, ideally with at least one marker staining (see Figure 2 from Wang et al., 2015). RNA-seq only measures the bulk expression; individual cells are not identifiable. This is especially pertinent since the TPM of NPY is low, and POMC and NPY are the two dominant neuron populations that are expected in this differentiation protocol [1]. Similarly, the testing for absence of FOXG1 in HP would be very informative to validate the input starting

material.

While the authors do not report any findings related to small non-coding RNAs, they technically include them in their reference genome (line 410, line 837). Since small non-coding RNAs are not efficiently captured in the standard RNA extraction and library prep methods that the authors describe in their method section, I would recommend to exclude them from analysis. Without special consideration during both extraction and library prep, the analysis of small non-coding RNAs is not reliable.

Maybe for Fig 4A a smaller font might make the labeling of phenotypes clearer. Also it would be interesting to include more disorders that are not expected to have hypothalamic involvement, as a negative control. It would also be interesting to hear the authors's thoughts on the observed enrichment of GWAS loci in cREs from undifferentiated ESC, since these should not be disease-specific, but have almost as many significant disease associations as the more relevant HP and HN cell types. Reassuringly, while they find MDD and BP loci enrichment in the hypothalamic samples, the strongest signals come from BMI, AAM, height, and sleep, which are the strongest hypothalamic phenotypes. Another open question that would be interesting to add as a small vignette is whether cREs contacting genes change in a direction that is consistent, i.e. more open regions show increased gene expression.

For clarity and context it would be helpful if the authors discussed the aspect of fetal brain development and the pathophysiology of body weight, height, onset of puberty, and other phenotypes controlled by the hypothalamus. I believe the Wang et al. paper compares the d29 in vitro HNs to E12.5-E18.5 neurons in mice with regards to overlapping expression patterns of POMC and NPY [1]. Do the authors intend to focus on prenatal neurons to decipher the epigenetics behind hypothalamus pathophysiology? Or is this something that should be discussed as limitation in the discussion section? Then it might fit in the section where they discuss mixed population of neurons present (line 271).

Further minor edits are:

The method description was comprehensive and illustrated the work done by the authors. It allowed the reader to follow along the work and its results. One minor thing I would ask for in addition is if the authors could describe what quality control they included for ATAC-seq samples.

Another small thing that is shown in Fig 4D, but which I would also mention in the text when describing the result, is that the SNP in the NRF1 motif is in a nucleotide position that is very variable.

Line 32: should yield -> could yield

Line 58: Should lead to greater understanding -> could lead to greater

Line 71: to can -> can

Line 93: the paper that they cite says differentiation efficiency is "80-95%" or "over 90%" [1]

Figure 1A: Meaning of "Differentiation-Function" Venn diagram not quite clear

Supp Fig 3B: y-axis?

Line 140: cRE abbreviation was not introduced in text yet. Add (cRE) in introduction section, line 66.

LI 786-787: "Gene expression and chromatin architecture underlies during hypothalamic neuron differentiation" is unclear.

Fig 3B: scale?

Are figure legends for Fig 4B and C are switched?

Line 272: wee -> were

1. Wang, L. et al. Differentiation of hypothalamic-like neurons from human pluripotent stem cells. *J. Clin. Invest.* 125, 796–808 (2015).
2. Inoue, F. et al. Genomic and epigenomic mapping of leptin-responsive neuronal populations involved in body weight regulation. *Nat Metab* 1, 475–484 (2019).
3. Willer, C. J. et al. Six new loci associated with body mass index highlight a neuronal influence on body weight regulation. *Nature Genetics* 41, 25–34 (2009).
4. Ong, K. K. et al. Genetic variation in LIN28B is associated with the timing of puberty. *Nature Genetics* 41, 729–733 (2009).
5. Cousminer, D. L. et al. Genome-wide association and longitudinal analyses reveal genetic loci linking pubertal height growth, pubertal timing and childhood adiposity. *Human Molecular Genetics* 22, 2735–2747 (2013).

POINT-BY-POINT RESPONSES TO REFEREE COMMENTS

We are most grateful for the opportunity to address the reviewers' comments and submit a revised manuscript. Point-by-point responses are presented below.

REVIEWER #1

"The manuscript is extremely well written, and in contrast to many other genomic papers, the logical links of the analysis are easy to follow and sound. Moreover, the authors clearly state results and their relevance but do not make attempts to oversell their data. Such genomic datasets from disease-relevant cell types are important to harvest the information obtained from large GWAS-studies and provide an important resource for further functional studies".

We are grateful to the reviewer for this positive feedback.

"Before publication, I suggest following minor changes to the text and figures to improve the readability of the study:

Minor points:

1. Despite having a considerable amount of whitespace, the authors avoid labeling individual tracks and displays in the Figures, but rather present all the information in the Figure legends. I strongly suggest adding some more information to the individual panel labels, as it is easier to understand for the reader. E.g. adding RNA-seq and ATAC-seq to the graphs in Fig. 1B/C and to the PCAs in Fig. 1a/b, adding differential accessibility or something similar to Fig. 2D. There could be more similar changes".

We thank the reviewer for this suggestion. We have now gone through the figures in order to add labels to panels to indicate where experiment the analysis was derived. Specifically, we added labels to Fig 1B.,C,D,E, Fig2C,D,E. Sup Fig 1A,B,E

2. "Figure 3a and Text: It seems as if the expression level of the respective TFs does not correlate so well with the Purity score or number of binding sites for the respective TF. This appears to be a limitation of the bioinformatic tool or purity of their cell populations. The authors should mention this".

The reviewer is correct that the TF analysis only showed a modest correlation with TF expression. While we agree it may be due to shortcomings in the bioinformatic tool, differences in quality of the position weight matrixes or diversity of cell types, it can also arise from posttranscriptional regulation. We added a paragraph to the Discussion addressing the TF results and caveats (line 303).

3. "Table 1: Explicitly clarify the column labeling in the manuscript text or rename: what are proxies? What are open proxies?"

Thanks for catching this omission. We have added this information in the Table 1 legend. (Table 1; Line 913).

4. “The authors should include an overview Supplemental Figure representing the interaction profile observed in the 3 different cell types. How many promoter-interacting regions did they find in each cell type and what was the mean number of interacting OCRs per gene?”

We now include a figure with the number of loops called per cell stage to supplemental figure 1. We have added the number of promoter contacts found in each cell type to line 153-155. The number of OCRs in contact are now presented in Figure 2B and the number of cRE per gene is now presented in Supplemental Figure 2C. The mean number of PIR-OCRs per cell type was ESC: 6.14, HP 4.70, and HN 4.70, the median of each cell type was 3. We have now added the median number to line 161.

5. “Although the heterogeneity in especially HN neurons is addressed in the discussion section, this could also be mentioned at the respective results section. This will help the reader to evaluate/interpret the presented data”.

We added a paragraph to the results section (starting on line 132) to describe some of the challenges associated with the iPSC model including addressing the cell heterogeneity. We highlight that the HNs express markers of several neuronal subtypes while some are lower/likely underrepresented.

REVIEWER #3

“While the approach of integrating several data forms into a single, coherent dataset is conceptually interesting, and the authors should be commended on assembling a clearly written manuscript with a nicely detailed materials and methods section”.

We thank the reviewer for the positive feedback.

“However, the study remains preliminary in its current state and would be improved if several major issues were addressed.

Major comments

1. The authors focus on human pluripotent stem cell (hPSC)-derived hypothalamic progenitors and young neurons as a model system. This is undeniably an interesting region of the brain to examine, but it is unclear which of their results might have been obtained by comparing hESCs to generic neural progenitors and generic young neurons, rather than hypothalamic progenitors and hypothalamic neurons specifically. Indeed, the author’s comparison to GTEx reveals a strong correlation of their cells to not just to hypothalamus but also to other brain regions (Fig. 1E)”.

It is true that the hESC derived HNs correlate strongly to neurons beside the hypothalamus. Comparisons across the GTEx data suggest that the different neuronal populations are highly correlated with one another and form an obvious cluster within GTEx data¹.

To supplement the comparison with GTEx, we used a database of brain region specific marker genes to compare the top 500 expressed genes expressed in HNs, which had

been previously used to verify assay the identity of iPSC derived neurons². Briefly, the curators of the database assigned a specificity score by taking expression data from different brain regions. We compared the top 500 genes expressed in the hypothalamus, and detected the strongest enrichment for genes associated with the hypothalamus hypocretineric neurons (Fisher's Exact test: FDR = 3.863×10^{-04}) as well as weaker enrichment detected in Striatal Cholinergic Neurons (Fisher's Exact test: FDR = 0.010). From this analysis we feel confident the neurons resemble the hypothalamus compared to other brain regions. We added this comparison to Supplemental Figure 1.

"In order to make conclusions about cREs relevant to hypothalamic cell function, the authors should perform similar analysis on hPSCs differentiated to other neuronal lineages (e.g. cortical, midbrain, spinal cord) for which robust differentiation protocols exist. Indeed, this comparison might give them greater power to reveal which cREs are hypothalamus-specific which would aid in the interpretation of GWAS coloc analysis for traits thought to be tied to the function of hypothalamic cells".

We thank the reviewer for the suggestion and agree that delving into differences between neuronal subtypes is an interesting research direction, however a comprehensive comparison of iPSC models of different brain regions falls outside the scope of our study. While multiple recent studies comparing promoter contacts in different neuronal and glial subsets in the context of neuropsychiatric disease, we are interested in investigating the potential impact of hypothalamic relevant GWAS SNPs using this model.

Our lab has recently generated a dataset on iPSC derived cortical-like neurons for a different study³. A direct comparison of these datasets would have many confounding factors (these cells were not generated from the same origin, sequenced in different batches, not prepared by the same individuals) given that this was not our initial goal, but the same protocols were used in library generation (in particular both were generated using Capture C using DpnII). We performed comparisons of the binary peak calls and the results for variant to gene mapping of the indicated traits. With the caveats mentioned, this analysis suggests that there are distinct results between different cell types.

2. "The authors identify only a modest number of colocalizations and appear to have selected GWAS based on the likely involvement of hypothalamic cells (e.g. growth/height, BMI, and sleep, although bipolar disorder, PTSD, and depression are also included) and are a mixture of well-powered studies, and less well-powered studies. In order to better interpret whether the author's data provides insights into the specific traits they wish to examine, it would be interesting to compare to other GWAS relevant to the nervous system but not hypothalamus (e.g. schizophrenia) and ask if there is any enrichment in colocs for hypothalamus-associated traits. If the authors they had data from hPSCs differentiated to other brain regions (see comment 1), this comparison would be even more valuable".

Our intention was to investigate the cREs for enrichment of hypothalamic relevant traits, given the hypothalamus's known role as a systemic regulator of metabolism/endocrinology/mood disorders. Thus, we feel that a comprehensive comparison with different neuropsychiatric disorders falls outwith of the scope of our study.

Our lab had generated IPS derived cortical-like neurons for another study, where we investigated the potential genetic enrichment for neuropsychiatric traits in IPS derived neural progenitors and neurons from a protocol expected to yield cortical like neurons³. We have now performed the requested comparisons for the neuropsychiatric traits in hypothalamic neurons. Specifically we checked for enrichment in GWAS studies for Attention Deficit Disorder, Autism, Anti-social disorder, Schizophrenia, and did indeed detect significant enrichment for ADHD (ESC: FDR=0.01, HP: FDR=0.008, HN: FDR=0.008), Schizophrenia (ESC: FDR=7.5x10⁻⁴, HP: FDR= 1.7x10⁻⁴, HN= FDR=1.2x10⁻³), and Intellectual disability (ESC= 1.8x10⁻³, HP=1.7x10⁻⁴, HN= 3.5x10⁻⁴). These results are indeed similar to what we observed in the cortical neurons, which suggests that there are elements in our data that are likely relevant to young neurons.

3. "One potentially useful application of the integrated dataset would be to leverage ATAC-seq and chromatin conformation capture data to expand the list of genes nominated by GWAS based on long-range interactions. For example, are there any examples of genes that might be affected by at a distance from GWAS loci and that have coding variants of large effect in traits relevant to the hypothalamus (e.g. height, BMI) in human datasets?"

This was performed on the GWAS that were enriched in the partitioned LD score regression analysis (Figures 4C, D, E, Supplemental Table 5). We now discuss several examples in the manuscript in the "Variant-to-gene mapping identifies target effector genes at GWAS-implicated loci" section.

4. "The authors focus on bulk methods for data collection yet the hypothalamus is developmentally complex composed of a multitude of individually rare cell populations. While it is true that single-cell methods for chromatin capture are limited, single-cell RNAseq and ATAC-seq are well-established and could provide additional value to their study to help interpret whether any co-localisation might be cell type dependent, and to measure heterogeneity in their samples to estimate which cell populations are contributing most to the described cREs. Furthermore, it's not clear why the specific time points in differentiation were selected and how closely these cell populations reflect those found in the adult. For example, how does the maturity of cells produced in this study compare to hypothalamic developmental datasets from the mouse (e.g. Kim et al, Nature Communications, 2020)? In my opinion, it's difficult to ascribe likely biological relevance to the results since the study has no comparison groups and is sampling a complex mix of cells whose relevance to the adult human hypothalamus is not clearly established".

With respect to the specific timepoints assayed, Wang et al. established the time course analysis for hypothalamic marker genes from distinct developmental stages. From a comprehensive assay for expression of hypothalamic differentiation markers, Wang et al. showed that day 12 of differentiation the cells express the hypothalamic progenitor marker NKX2-1 as well as the general neuroprogenitor marker Nestin. At day 12 the neuronal marker TUBB3 is expressed at low levels and thus, closely resemble a progenitor phenotype. At day 27 the neurons reach maximum expression of the neural markers TUBB3 and POMC. And therefore, day 27 was chosen as the time point of collection. We have now added a sentence on line 97 clarifying this point.

We do acknowledge there are limitations by using pluripotent stem cell derived neurons, particularly that they will not fully resemble *in vivo* mature neurons that would be found in adulthood, as both iPSC and organoid models have difficulty reaching terminal stages of neuronal differentiation⁴. However, the neurons generated by this protocol have been shown to: express molecular markers consistent with hypothalamic identity, be electrophysiologically active, respond to hormones such as insulin & leptin⁵. Thus, we believe these neurons to be a valuable cellular model for the rare human hypothalamic neurons and profiling their chromatin architecture provides a valuable resource to integrate with GWAS data for future function investigations.

5. “The authors appear to have performed the study using three distinct differentiations of a single cell line (H9 or WA09) at only two time points in the differentiation process. Given the well-known variability in differentiation propensity between cell lines, biological replicates should be generated using at least 3 distinct genetic backgrounds. Otherwise, conclusions should be drawn more narrowly and discussed in the context of the WA09 cell line at the specific differentiation time points tested”.

We have added caveats early one in the results section (Lines 93-94). We selected the H9 line because it is a widely used stem cell line for neurodifferentiation and thus, a well-accepted cellular model^{6,7}.

6. “The authors demonstrate modest success of confirming that some known GWAS hits are co-localised with cREs, but the reader is left unsure which of the results are sufficiently compelling to warrant future studies. Any further validation of their by an orthogonal method or demonstration of relevance to the co-localised traits would greatly increase the potential impact of the study”.

We attempted co-localization analysis of the GWAS reports with eQTLs with limited success. We reasoned that incorporating chromatin conformation data would point to genes with known monogenetic mutations that affect our cell types of interest (biologically consistent with rare disorders that likely have larger effect on the gene). We found a SNP located in an intron of *CADPS2* that contacted the promoter of *FEZF1*. Mutations in *FEZF1* are associated with rare mendelian disorders that are associated with delayed/absent puberty.

We observed several other genes with known to influence rare monogenetic mutations, and we now highlight several additional in Lines 298-299 and 304-305.

“Minor comments

1. The term cRE is not defined at first mention (line 140) and is sometimes inconsistently called cisRE (line 802, Fig S3)”

We corrected this error on lines 67, 802 and in Figure S3.

2. “The discussion of *FEZF1* and its association with Kallman syndrome seems like a stretch, since any cRE are likely to modestly affect *FEZF1* expression and Kallman syndrome is an autosomal recessive disease requiring severe impairment of gene function”.

We were not implying that this SNP itself is responsible Kallman syndrome, but were using this example to provide evidence that the gene targeted by the SNP points to a gene with orthogonal support in the context of age at menarche. Rare loss of function mutations in *FEZF1* are known to result in Kallman Syndrome, which suggests that *FEZF1* is a puberty gene⁸. The highlighted SNP is predicted to have a small effect on the TF binding site is consistent with the notion that it may have a small contribution on a developmentally relevant gene.

3. “In Figure 1D, it is difficult to see that there is a difference in the number of interaction arcs, and which of these changes are significant. Also, the colour scheme of HP and HN in 1D is inconsistent with that in 1A and 3D,E”.

We thank the reviewer for catching this error. We have now corrected the color scheme for the panel. We have also added the number of interactions for each celltype.

4. “In Figure 2C, it is difficult to distinguish between those peaks that are dashed (for PIR-OCR) and those that are not (for non PIR-OCR). Additionally, Figure 2D could be made more clear by the addition of cell type transition labels, similar to those in Figure 2E”.

We thank the reviewer for catching this. We have now made the dash marks more visible.

5. “In the legend for Figure 4, the description for panels B and C appears to have been swapped”.

We have now corrected the figure legend.

REVIEWER #4

“In this study the authors analyzed cis-regulatory architecture in an in vitro model of hypothalamus embryonic and fetal development. The authors used a human embryonic stem cell line and applied a published protocol for hypothalamic neuron differentiation [1]. This generated a hypothalamic progenitor (HP) cell sample and a hypothalamic neuron (HN) cell sample. These samples were studied using RNA-seq, ATAC-seq and Capture C. The identified loci were cross-referenced with published GWAS SNPs from studies on phenotypes controlled by hypothalamus function, for example body weight, height, and onset of puberty.

The authors were able to find genes that have a known function in these phenotypes. This shows the utility of in vitro models - in the absence of primary tissues - to study regulatory networks. It also shows the strength of using several levels of genomic information, such as gene expression, chromatin openness and conformation, as well as SNP associations. While the in vitro models have inherent shortcomings and represent a simplified model of neuronal cell, they have the advantage of being more amenable to molecular studies.

The major strength of this paper lies in showing that in vitro models do have some usefulness. As the authors show, significant signals can be detected that can be connected to the hypothalamus and its function. The strength lies in the availability of starting material, which allowed execution of these analyses. Another strength of the paper is the combination of

different layers of information, for example cross-referencing cis-regulatory regions with GWAS signals. Studies of this kind are very timely and useful to the field of genetics of metabolism and endocrine signaling”.

We greatly appreciate these positive comments from the reviewer.

“A few changes could improve the manuscript even further. For example, Inoue et al. published a study in 2019 studying the epigenome of leptin-responsive genes in the mouse hypothalamus [2]. While this study did not look at human tissue, but mouse tissue, it would still be interesting to compare the identified loci, since it is the only study to my knowledge that also looked at the epigenetics and not only gene expression of the hypothalamus. If there is an overlap between loci, this would strengthen the findings further. It would also be interesting to see if the same loci are implicated in development (present manuscript) and young adult stage (Inoue)”.

We thank the reviewer for the suggestion. We performed this comparison and have now added a new Supplemental Figure 4 with the results. The Inoue 2019 open ATAC-seq/H3K27ac ChIP-seq peaks region calls were converted to the homologous human coordinate’s peaks. We observed a significantly enriched overlap our set of peaks and their peaks.

“Other published studies that could be referenced in the manuscript are Willer et al., Nat Genet 2009; Ong et al., Nat Genet 2009; and [3,4,5]. These would fit well in line 60-61 when the authors cite other GWA studies. The currently referenced studies center around sleep and chronotype. I believe it would be fitting to also reference GWA studies related to other phenotypes regulated by hypothalamus, such as body weight and puberty”.

We thank the reviewer for catching this omission. We have now added references Willer et al., Ong et al., and Cousminer et al. to line 60.

“Especially since the authors study the hypothalamus, which as they say controls features such as reproductive timing and body weight. Those features show sex differences, which raises the question of whether the experimental design should have included one XX and one XY cell line. While this cannot be changed now without repeating the entire study, it is worth considering whether to make it clearer for the reader in the manuscript that results come from one XX cell line and its potential limitations:”

This is an important point and a limitation of our study. We have now added a paragraph to highlight this caveat in the discussion (lines 354-365).

“Also, with regards to the cellular samples that are the input material into the genomics assays and the basis of all analyses, more information about the starting material would strengthen the manuscript. Supp Fig 1E shows RNA expression of some marker genes as quality control, but since they are basically studying a cell model, it would be nice to see a figure with images of the cells, ideally with at least one marker staining (see Figure 2 from Wang et al., 2015).

RNA-seq only measures the bulk expression; individual cells are not identifiable. This is especially pertinent since the TPM of NPY is low, and POMC and NPY are the two dominant neuron populations that are expected in this differentiation protocol [1].

Similarly, the testing for absence of FOXP1 in HP would be very informative to validate the input starting material”.

We agree with the reviewer on adding markers profiling the differentiation of HNs. To that end, we have now added images of HNs labeled for POMC and MAP2 to Supplemental Figure 1. For FOXP1 TPM values from the RNA-seq to the set of marker genes displayed on Supplemental Figure 1E. We observed very low levels of FOXP1 expression in HPs, while rising later in HNs. This may correspond to some FOXP1+ neurons that appear later in hypothalamic development.

“While the authors do not report any findings related to small non-coding RNAs, they technically include them in their reference genome (line 410, line 837). Since small non-coding RNAs are not efficiently captured in the standard RNA extraction and library prep methods that the authors describe in their method section, I would recommend to exclude them from analysis. Without special consideration during both extraction and library prep, the analysis of small non-coding RNAs is not reliable”.

We thank the reviewer for catching this, it is correct that the estimates for small non-coding RNA expression levels will not be accurate, so we have removed these from the supplemental tables 4,5,7. We will also contact GEO to adjust the uploaded processed RNA-seq data.

“Maybe for Fig 4A a smaller font might make the labeling of phenotypes clearer. Also it would be interesting to include more disorders that are not expected to have hypothalamic involvement, as a negative control. It would also be interesting to hear the authors’s thoughts on the observed enrichment of GWAS loci in cREs from undifferentiated ESC, since these should not be disease-specific, but have almost as many significant disease associations as the more relevant HP and HN cell types.

Reassuringly, while they find MDD and BP loci enrichment in the hypothalamic samples, the strongest signals come from BMI, AAM, height, and sleep, which are the strongest hypothalamic phenotypes”.

We agree that enrichment and many more genes were identified by variant to gene mapping. We reduced the font size and replaced the acronyms with the GWAS trait name. We also removed the abbreviations from the legend (838-842).

For partitioned LD score regression, we did not use the cREs that were specific to each celltype, we used all cREs with FPKM > 1 in each celltype (which were treated as binary calls). Given that there are more more cRE per gene in ESCs it’s possible that it’s driven by shared cREs to those cells.

As requested we have now added two negative control traits, type 1 diabetes and lumbar spine bone mineral density.

“Another open question that would be interesting to add as a small vignette is whether cREs contacting genes change in a direction that is consistent, i.e. more open regions show increased gene expression”.

Figure 2 shows that the overall pattern of accessibility doesn't match the expression pattern during development (that globally most sites open from ESC>HP and most sites close from HP > HN regardless of direction of how the cluster). We have now tested whether the changes in the number promoter contacts to OCRs correlates with changes in expression. We did not observe a strong correlation.

“For clarity and context it would be helpful if the authors discussed the aspect of fetal brain development and the pathophysiology of body weight, height, onset of puberty, and other phenotypes controlled by the hypothalamus. I believe the Wang et al. paper compares the d29 in vitro HNs to E12.5-E18.5 neurons in mice with regards to overlapping expression patterns of POMC and NPY [1]. Do the authors intend to focus on prenatal neurons to decipher the epigenetics behind hypothalamus pathophysiology? Or is this something that should be discussed as limitation in the discussion section? Then it might fit in the section where they discuss mixed population of neurons present (line 271)”.

While we focused on the how GWAS SNPs may act during development by comparing the different cellular stages, it is indeed a limitation that neurons do not reach full maturity. We have included this caveat in lines 367-380.

“Further minor edits are:

The method description was comprehensive and illustrated the work done by the authors. It allowed the reader to follow along the work and its results. One minor thing I would ask for in addition is if the authors could describe what quality control they included for ATAC-seq samples”.

Similar to RNA-seq we verified groups by principal component analysis and Pearson correlation plots (Supplemental Figure 1). In addition, we verified the presence of mono-nucleosome and di-nucleosome in fragment distribution plots. We have now added this to the methods line 575.

“Another small thing that is shown in Fig 4D, but which I would also mention in the text when describing the result, is that the SNP in the NRF1 motif is in a nucleotide position that is very variable”.

We have now added this caveat to line 260.

“Line 32: should yield -> could yield”

We thank the reviewer for these corrections. We have now changed “should yield” to “could yield”

“Line 58: Should lead to greater understanding -> could lead to greater”

We have now changed “should” to “could” lead

“Line 71: to can -> can”

We have removed “to”

“Line 93: the paper that they cite says differentiation efficiency is “80-95%” or “over 90%” [1]”

We have now revised the sentence to correct the number to 80-95%

“Figure 1A: Meaning of “Differentiation-Function” Venn diagram not quite clear”

We were intending to summarize the outcome of the study (identifying some SNPs that act during development). We have now modified the summary diagram removing the Venn Diagram.

“Supp Fig 3B: y-axis?”

We have now corrected the axis from “# cisRE” to “# contacting cRE per gene”

“Line 140: cRE abbreviation was not introduced in text yet. Add (cRE) in introduction section, line 66”.

We have now added the earlier cRE abbreviation on line 67.

“LI 786-787: “Gene expression and chromatin architecture underlies during hypothalamic neuron differentiation” is unclear”.

We have corrected the figure title to “Gene expression and chromatin architecture dynamics of gene during hypothalamic neuron differentiation”

“Fig 3B: scale?”

We have added the scale bar back to the image.

“Are figure legends for Fig 4B and C are switched?”

Thank you for catching this mistake. We have now swapped the figure legend (lines 962-966).

“Line 272: wee -> were”

Thank you, we have now corrected this typo.

References:

1. Consortium GT, *et al.* Genetic effects on gene expression across human tissues. *Nature* **550**, 204-213 (2017).
2. Rajamani U, *et al.* Super-Obese Patient-Derived iPSC Hypothalamic Neurons Exhibit Obesogenic Signatures and Hormone Responses. *Cell Stem Cell* **22**, 698-712 e699 (2018).
3. Su C, *et al.* 3D promoter architecture re-organization during iPSC-derived neuronal cell differentiation implicates target genes for neurodevelopmental disorders. *Prog Neurobiol*, 102000 (2021).
4. Bhaduri A, *et al.* Cell stress in cortical organoids impairs molecular subtype specification. *Nature* **578**, 142-148 (2020).
5. Wang L, Egli D, Leibel RL. Efficient Generation of Hypothalamic Neurons from Human Pluripotent Stem Cells. *Curr Protoc Hum Genet* **90**, 21 25 21-21 25 14 (2016).
6. Besusso D, *et al.* A CRISPR-strategy for the generation of a detectable fluorescent hESC reporter line (WAe009-A-37) for the subpallial determinant GSX2. *Stem Cell Res* **49**, 102016 (2020).
7. Kriks S, *et al.* Dopamine neurons derived from human ES cells efficiently engraft in animal models of Parkinson's disease. *Nature* **480**, 547-551 (2011).
8. Kotan LD, *et al.* Mutations in FEZF1 cause Kallmann syndrome. *Am J Hum Genet* **95**, 326-331 (2014).

REVIEWER COMMENTS

Reviewer #3 (Remarks to the Author):

I thank the authors for the time and effort they have taken to address the reviewers' comments, which have indeed strengthened the manuscript. However, the primary remaining concern in my view is that the breadth and reach of the claims are not fully supported by the data provided. The conclusions depend on a single cell lineage (HP and HN) derived from a single cell line (even if it is widely-used) with neurons collected at an early time point (day 27) when they are likely to still be immature. Human stem cell-derived cellular models are powerful, but fundamentally limited in the purify and maturity of the cell populations they can yield. I believe it is essential for the authors to either a) provide additional data to enable them to make stronger claims, or b) clarify that any claims made can only be evaluated within the context of their specific model system and data sets. I appreciate that it is costly and time-consuming to generate additional data, and the authors correctly state that it is problematic to cross-compare with other datasets. However, stating that it is "outside of the scope of our study" to compare hypothalamic cells to those from at least one other brain region differentiated from the same cell line indicates that the study was not designed to enable the authors make any claims about cREs specific to or relevant in human hypothalamic neurons, nor to enable the authors to suggest that the GWAS associations might be relevant for human to disease when present in their dataset, especially if GWAS traits were examined selectively (I do appreciate that a handful of negative controls are now included). I'm sure that the authors would not wish readers to be misled, so I suggest that they clearly state caveats to the experimental design alongside reporting of the results, and also devote some time to this in the discussion, perhaps as part of the nice discussion of caveats in lines 358-371.

Reviewer #4 (Remarks to the Author):

The authors addressed suggestions sufficiently.

Minor corrections:

Figure legend for Supp Fig 4D missing.

REVIEWER COMMENTS

Reviewer #3 (Remarks to the Author):

I thank the authors for the time and effort they have taken to address the reviewers' comments, which have indeed strengthened the manuscript. However, the primary remaining concern in my view is that the breadth and reach of the claims are not fully supported by the data provided. The conclusions depend on a single cell lineage (HP and HN) derived from a single cell line (even if it is widely-used) with neurons collected at an early time point (day 27) when they are likely to still be immature. Human stem cell-derived cellular models are powerful, but fundamentally limited in the purity and maturity of the cell populations they can yield. I believe it is essential for the authors to either a) provide additional data to enable them to make stronger claims, or b) clarify that any claims made can only be evaluated within the context of their specific model system and data sets.

I appreciate that it is costly and time-consuming to generate additional data, and the authors correctly state that it is problematic to cross-compare with other datasets. However, stating that it is "outside of the scope of our study" to compare hypothalamic cells to those from at least one other brain region differentiated from the same cell line indicates that the study was not designed to enable the authors make any claims about cREs specific to or relevant in human hypothalamic neurons, nor to enable the authors to suggest that the GWAS associations might be relevant for human to disease when present in their dataset, especially if GWAS traits were examined selectively (I do appreciate that a handful of negative controls are now included). I'm sure that the authors would not wish readers to be misled, so I suggest that they clearly state caveats to the experimental design alongside reporting of the results, and also devote some time to this in the discussion, perhaps as part of the nice discussion of caveats in lines 358-371.

As requested we included additional caveats in the results (lines 215, 249-253) and in the discussion (lines 366-369). For the latter, we have specifically stated: "Another limitation in our experimental design is that we examined neurons generated to resemble a single brain region, which limits our ability to distinguish cREs that may be shared across cell types or specific to a hypothalamic context. Due to this limitation, it is likely that some GWAS signals intersecting HP/HN cREs may be common to neural progenitors/young neurons from multiple brain regions or not represented *in vivo* hypothalamic neurons."

Reviewer #4 (Remarks to the Author):

The authors addressed suggestions sufficiently.

We thank the reviewer for the input that has strengthened the manuscript.

Minor corrections:

Figure legend for Supp Fig 4D missing.

We have corrected the legend.

REVIEWERS' COMMENTS

Reviewer #3 (Remarks to the Author):

The two sentences added to the discussion do indeed point out a limitation of the study to the reader, but there are still claims made throughout the study are not fully supported by the evidence. In my view, language should be used more carefully throughout the manuscript if additional confirmatory data is not forthcoming. For example, before a cell population can be called "hypothalamic neurons" it seems incumbent on the authors to back up this claim by either 1) showing the 95% purity of the neuronal composition they claim these populations to be among the analysed samples by quantified immunostaining or single-cell sequencing data, 2) using a more accurate term such as "hypothalamic cells" if cell composition data is lacking, or 3) acknowledging up front the heterogeneity and limitation of the method. For example they might say that: the term "hypothalamic neurons (HN)" will be used to describe this cell population composed of a diverse population of hypothalamic neurons, and a small population of non-neuronal cells.

I like this study, but think that if minor textual modifications such as this were added throughout the manuscript (not just briefly mentioned in the discussion) that it would avoid misinterpretations by non-experts and therefore be more valuable to broad readership of Nature Communications.

REVIEWERS' COMMENTS

Reviewer #3 (Remarks to the Author):

The two sentences added to the discussion do indeed point out a limitation of the study to the reader, but there are still claims made throughout the study are not fully supported by the evidence. In my view, language should be used more carefully throughout the manuscript if additional confirmatory data is not forthcoming. For example, before a cell population can be called "hypothalamic neurons" it seems incumbent on the authors to back up this claim by either 1) showing the 95% purity of the neuronal composition they claim these populations to be among the analysed samples by quantified immunostaining or single-cell sequencing data, 2) using a more accurate term such as "hypothalamic cells" if cell composition data is lacking, or 3) acknowledging up front the heterogeneity and limitation of the method. For example they might say that: the term "hypothalamic neurons (HN)" will be used to describe this cell population composed of a diverse population of hypothalamic neurons, and a small population of non-neuronal cells.

I like this study, but think that if minor textual modifications such as this were added throughout the manuscript (not just briefly mentioned in the discussion) that it would avoid misinterpretations by non-experts and therefore be more valuable to broad readership of Nature Communications.

We have additional acknowledgements of the limitations of our study. In addition to the modifications to the discussion, we have also added additional caveats to the last paragraph of the introduction (lines 87-90). "While the hypothalamus consists of a diverse array of neuronal subtypes, we approached this using bulk sequencing approach on differentiated cells. The term "hypothalamic neurons (HN)" will be used to describe the differentiated cell population composed of a diverse set of differentiated hypothalamic-like neurons, and a small population of non-neuronal cells."